

# Physical Properties of Secondary Photochemical Aerosol from OH Oxidation of a Cyclic Siloxane

Nathan J. Janechek[1,2], Rachel F. Marek[2], Nathan Bryngelson[1,2,*], Ashish Singh[1,2,†], Robert L. Bullard[1,2,‡],William H. Brune[3], Charles O. Stanier[1,2]

[1]Department of Chemical and Biochemical Engineering, University of Iowa, Iowa City, IA, USA
[2]IIHR-Hydroscience and Engineering, University of Iowa, Iowa City, IA, USA
[3]Department of Meteorology and Atmospheric Science, Pennsylvania State University, University Park, PA, USA
[*]Now at Yokogawa Corporation of America - Analytical Process Analyzers
[†]Now at Institute for Advanced Sustainability Studies, Potsdam-14467, Germany
[‡]Now at Lovelace Respiratory Research Institute, Albuquerque, NM, USA

*Correspondence to*: Charles O. Stanier (charles-stanier@uiowa.edu)

**Abstract.** Cyclic volatile methyl siloxanes (cVMS) are high production chemicals present in many personal care products. They are volatile, hydrophobic, and relatively long-lived due to slow oxidation kinetics. Evidence from chamber and ambient studies indicates that oxidation products may be found in the condensed aerosol phase. In this work, we use an oxidation flow reactor to produce ~100 µg m$^{-3}$ of organosilicon aerosol from OH oxidation of decamethylcyclopentasiloxane (D$_5$) with aerosol mass fractions (i.e. yields) of 0.2-0.5. The aerosols were assessed for concentration, size distribution, morphology, sensitivity to seed aerosol, hygroscopicity, volatility and chemical composition through a combination of aerosol size distribution measurement, tandem differential mobility analysis, and electron microscopy. Similar aerosols were produced when vapor from solid antiperspirant or from hair conditioner was used as the reaction precursor. Aerosol yield was sensitive to chamber OH, indicating an interplay between oxidation conditions and the concentration of lower volatility species. The D$_5$ oxidation aerosol products were relatively non-hygroscopic, with average hygroscopicity kappa of ~0.01, and nearly non-volatile up to 190°C temperature. Recommended parameters for treatment as a semi-volatile organic aerosol in atmospheric models are provided.

## 1 Introduction

Cyclic volatile methyl siloxanes (cVMS) are high production chemicals (OECD Environment Directorate, 2004) present in many personal care products such as lotions, hair conditioners, and antiperspirants (Horii and Kannan, 2008; Wang et al., 2009; Lu et al., 2011; Dudzina et al., 2014; Capela et al., 2016). Cyclic siloxanes are volatile (Lei et al., 2010), relatively unreactive, and hydrophobic molecules (Varaprath et al., 1996) composed of a Si-O ring backbone with two methyl groups bonded to each Si. The most prevalent cVMS in personal care products is decamethylcyclopentasiloxane (D$_5$) (Horii and Kannan, 2008; Dudzina et al., 2014; Lu et al., 2011; Wang et al., 2009). Cyclic siloxanes are readily released into the environment, primarily to the atmosphere (Mackay et al., 2015) by personal care product usage (Tang et al., 2015; Gouin et



al., 2013; Montemayor et al., 2013; Coggon et al., 2018). Once in the atmosphere, the primary environmental fate for cVMS is oxidation. The principle sink is reaction with the hydroxyl radical (OH) and the characteristic atmospheric lifetime is ~5-10 days (Atkinson, 1991). Measurements (McLachlan et al., 2010; Genualdi et al., 2011; Yucuis et al., 2013; Ahrens et al., 2014; Companioni-Damas et al., 2014; Tang et al., 2015) and modeling studies (McLachlan et al., 2010; MacLeod et al.,

2011; Xu and Wania, 2013; Janechek et al., 2017) show cVMS are ubiquitous. Maximum concentrations occur in indoor spaces and automobile cabin air, up to 380 µg m$^{-3}$ (Coggon et al., 2018) with outdoor urban locations up to 0.65 µg m$^{-3}$ (Buser et al., 2013).

Personal care products are increasingly recognized as relevant sources of urban air pollution in developed cities (McDonald et al., 2018), are likely underrepresented in official inventories, and lead to poorly characterized ozone and

secondary organic aerosol production.

Cyclic siloxanes have undergone continuing regulatory scrutiny (ECHA, 2018) for environmental impacts such as persistence, bioaccumulation, and toxicity but have entirely focused on the parent (e.g. $D_5$) compounds, and have not considered oxidation products (Environment Canada and Health Canada, 2008c, a, b; Brooke et al., 2009b, a, c; ECHA, 2015). However, numerous studies have reported particle formation from cVMS oxidation, with a direct link in lab studies or

inferred in ambient studies (Bzdek et al., 2014). Chamber studies have shown that a range of non-volatile and semi-volatile oxidation products form upon reaction with OH that can form secondary aerosol (Latimer et al., 1998; Sommerlade et al., 1993; Chandramouli and Kamens, 2001; Wu and Johnston, 2016, 2017). Wu and Johnston (2017) studied the molecular composition and formation pathways of aerosol phase $D_5$ oxidation products at aerosol loadings of 1-12 µg m$^{-3}$ and identified three main types of species: monomer (substituted $D_5$), dimer, and ring-opened oxidation products.

Janechek et al. (2017) atmospheric modeling at 36 km resolution provided the first spatial distribution and potential loadings of the oxidation products, of which some fraction likely form aerosol species. Peak oxidized $D_5$ (o-$D_5$) occurs downwind of urban areas with a monthly average concentration of 9 ng m$^{-3}$. Cyclic siloxane oxidation may represent an important source of ambient secondary aerosols with health and climate implications due to potential high loadings and widespread use of the precursor compounds.

To improve the community's ability to understand the health and climate impacts of secondary organic silicon particle formation, additional physical and chemical details on the particles are required. In this work, aerosols are produced using an oxidation flow reactor (OFR) which are photochemical flow-through reactors. OFRs are small (liter sized) reactors that use very high oxidation conditions to oxidize reactants in a matter of minutes (Kang et al., 2007; Kang et al., 2011; Lambe et al., 2011a). Advantages of using OFRs include the ability to quickly test a range of aging conditions, reach high

oxidant exposures (multi-day exposure) in minutes, and deploy easily for ambient measurements. Additionally, OFRs have been successfully used to generate secondary organic aerosol (SOA) for measurement of hygroscopicity and for the impact of oxidative aging on cloud condensation nuclei (CCN) activity (Lambe et al., 2011a; Lambe et al., 2011b; Palm et al., 2018).



Due to high OH concentrations and higher energy UV light in OFRs, the applicability of generated aerosols under these conditions has been evaluated by comparing aerosol yield (aerosol mass produced / mass of reacted gas precursor) and aerosol composition to well-established environmental chambers. Previous work has generally concluded SOA yield and chemical composition are comparable between OFRs and environmental chambers (Kang et al., 2007; Kang et al., 2011; Lambe et al., 2011a; Lambe et al., 2011b; Lambe et al., 2015; Bruns et al., 2015; Chhabra et al., 2015; Palm et al., 2018). Lambe et al. (2015) reported aerosol composition was independent of the oxidation technique (OFR vs environmental chamber) for similar OH exposures. Slight differences in SOA yield have been reported with OFR yields both higher (Kang et al., 2007) and lower (Lambe et al., 2011a; Bruns et al., 2015; Lambe et al., 2015). Possible differences in yields have been discussed including uncertainty (Bruns et al., 2015), non-identical reactor conditions (Kang et al., 2007; Lambe et al., 2015; Bruns et al., 2015), fragmentation at high oxidant conditions (Kroll et al., 2009; Lambe et al., 2012), and differences in wall losses (Lambe et al., 2011a; Bruns et al., 2015). Wall losses are important to consider for both environmental and OFR chambers when performing yield measurements (Krechmer et al., 2016; Pagonis et al., 2017; Palm et al., 2016).

OFRs can be erroneously operated under conditions that favor non-OH reactions depending on the chamber conditions. A number of studies have been performed to model the reactor chemistry and develop best operating conditions so that reaction with OH is favored and atmospherically relevant (Li et al., 2015; Peng et al., 2016; Peng et al., 2015). Best operating conditions were recommended based on studying the relative influence of 185 nm and 254 nm photolysis, reaction by ozone ($O_3$), and $O(^1D)$ and $O(^3P)$ radicals. Additionally, Palm et al. (2016) developed a model to correct for non-aerosol condensation pathways of low-volatility vapor in the chamber. These alternative pathways are condensation to the chamber walls, fragmentation to non-condensable vapors, and residence times not long enough for condensation to occur. If attention is paid to these OFR chamber issues, atmospherically relevant oxidation conditions can be produced.

In this work, we generated and characterized secondary aerosol from oxidation of $D_5$ and personal care product precursors using an OFR. This study characterized the particles for concentration, size, morphology, and elemental composition by energy-dispersive X-ray spectroscopy. Aerosol mass formation (yield) was quantified and sensitivity of precursor concentration, oxidant exposure, residence time, and seed aerosol was studied. Particle hygroscopicity or cloud seed formation potential was characterized by measurement of the hygroscopicity kappa parameter ($\kappa$), which is a measure of the particle water-uptake (Petters and Kreidenweis, 2007). Additionally, particle volatility was assessed by exposing generated aerosols to high temperatures. Finally, aerosol formation by thermal degradation of the parent compound was explored to account for potential aerosol production after contact of $D_5$ with heated surfaces.

## 2 Methods

Fifteen experiments were performed (Table S1) which contained five experiments to test aerosol yield sensitivity to system parameters, an experiment to test sensitivity to seed aerosols, two $D_5$ gas quantification quality control tests, three tests with generation of aerosol from the off-gassing of personal care products, two hygroscopicity measurements, aerosol volatility



measurement, and thermal degradation of parent $D_5$. The quality control tests were breakthrough testing of $D_5$ on the collection cartridge and verification that photo oxidation was the cause of $D_5$ depletion.

## 2.1 Aerosol generation and characterization

A 13.3 L Potential Aerosol Mass (PAM) OFR chamber (Kang et al., 2007; Kang et al., 2011; Lambe et al., 2011a) was used to oxidize vapor phase $D_5$. The OFR was run in the OFR185 mode (Peng et al., 2015) where $O_3$ and OH are generated in situ. The chamber is designed to operate at very high oxidation conditions yet still maintain atmospherically relevant ratios of $OH/O_3$ and $HO_2/OH$ (Kang et al., 2007). All characterization and sampling was performed on the centerline exhaust while the 1 or 3 L min$^{-1}$ ring flow (with more exposure to the reactor wall) was discarded as shown in Fig. 1. Compressed air (3.5 or 5 L min$^{-1}$) was passed through activated carbon and HEPA filters, and humidified to 25-40% RH (Sigma-Aldrich Optima or Barnstead Nanopure filtered ultrapure reagent grade water). $D_5$ was introduced into the flow by diffusion from a heated Teflon tubing leg filled with liquid $D_5$ (Sigma-Aldrich, purity 97%) controlled using a water bath.

Alternatively, precursor gases were introduced to the system flowing air (5 L min$^{-1}$, unheated) past personal care products in an Erlenmeyer flask. Results from antiperspirant containing cyclopentasiloxane and leave-in conditioner containing cyclomethicone are reported herein. Prior to introduction to the system, the chamber was cleaned for ~24 h with the lights on and no precursor gas.

To remove organic gases and high concentrations of $O_3$ from the OFR effluent, two annular denuders using activated carbon (Fisher Scientific; 6x14 mesh size) and Carulite 200 (manganese dioxide/copper oxide catalyst; Carus Corp.) were used. The activated carbon denuder dimensions were 25 cm outside diameter (OD), 20 cm inside diameter (ID) by 125 cm, while the Carulite denuder had dimensions of 14 cm OD, 1 cm ID by 70 cm. The denuders were packed with material between the OD and ID. Teflon tubing was used upstream of the reactor and copper downstream. Silicon conductive tubing was minimized to limit silicon introduction from sources other than the main reagent.

Aerosols were measured using a TSI 3936L85 SMPS (TSI 3785 CPC, TSI 3080 classifier, TSI 3081 long column DMA, and TSI 3077 Kr-85 2mCi neutralizer). Mass concentration was estimated from the SMPS measured volume concentration assumed for spherical particles using a liquid $D_5$ density of 0.959 g cm$^{-3}$. Aerosol sampling covered 9.7 – 422 nm with a sheath flow of 6 L min$^{-1}$ and aerosol flow of 1 L min$^{-1}$. SMPS measurements were corrected for size specific particle losses from gravitational settling, diffusion, and turbulent inertial deposition caused by tubing, bends, and constrictions (Willeke and Baron, 1993).

Particles were collected on a carbon film nickel transmission electron microcopy (TEM) grids (SPI 200 mesh) for microscopy and elemental analysis using an RJ Lee Group, Inc. Thermophoretic Personal Sampler (TPS100, RJ Lee Group) (Leith et al., 2014). The TPS100 samples was were collected using a hot and cold surface temperature of 110°C and 25°C, respectively, flow rate of 0.005 L min$^{-1}$, and a 25-min sampling time. The TPS100 samples were analyzed using a field emission scanning electron microscope (FESEM) with scanning transmission electron microscopy (STEM) capabilities



(Hitachi S-5500). The FESEM was equipped with an energy dispersive X-ray spectroscopy (EDS) system incorporating a 30 mm$^2$ silicon drift detector (Bruker Quantax) for collection of elemental composition.

## 2.2 Yield

Five experiments were run to test the sensitivity of aerosol yield to residence time, OH exposure, and reactant concentration. The system settings varied were UV light intensity, flowrate, relative humidity, and $D_5$ water bath temperature. The influence of seed aerosols was quantified separately using 50 nm ammonium sulfate aerosols.

Experiments included a ~20 h equilibration period followed by an analysis period (~1.7 h) where the $D_5$ gas concentration was quantified upstream and downstream of the OFR. Gas samples were collected on solid phase extraction (SPE) cartridges. Immediately following $D_5$ gas collection, the precursor gas was switched to $SO_2$, allowing calculation of OH exposure. $SO_2$ with the OFR lights on was measured (Teledyne 100E) for an average of 4.6 h, followed by 3.3 h with the lights off. In this work, $SO_2$ loadings were 5 – 28 times higher than $D_5$. We recognize for the best estimate of OH exposure, similar concentrations are desired. The aerosol yield was calculated from the SMPS particle loss-corrected mass concentration and the reacted $D_5$ gas measurements (aerosol mass / reacted $D_5$ gas concentration).

### 2.2.1 OH exposure quantification

OFR OH exposure was quantified using two methods, comparison of $SO_2$ gas concentrations with and without UV light, and aerosol production from $SO_2$ oxidation as measured by SMPS. Reference $SO_2$ gas (5 ppm in $N_2$) was diluted using mass flow controllers with the volumetric flow verified using a bubble flow meter (Sensidyne Gilian Gilibrator-2). The $SO_2$ monitor was calibrated following the calibration procedure provided by the manufacturer (slope = 1.400, offset = 41.4 mV). The monitor calibration was checked using 337 and 981 ppb $SO_2$ reference gas with the measured monitor concentration within 1.8 and 1.4%, respectively.

The main method of OH exposure quantification used in the analysis was by the measurement of the disappearance of $SO_2$ in the OFR (Eq. (1)). The initial $SO_2$ gas concentration (450 – 1200 ppb) was measured at the OFR exit without UV illumination. The final gas concentration (6 – 190 ppb) was measured similarly, but with UV illumination. All analyzed gas concentrations were the last 20-min average for each respective period after equilibrating for 2-5 h. In Eq. (1) [OH]*t represents the OH exposure (molec s cm$^{-3}$), [$SO_2$] and [$SO_2$]$_0$ the final and initial gas concentrations, respectively, and $k_{SO_2}$ the OH rate constant. The $SO_2$ oxidation rate used was 9x10$^{-13}$ cm$^3$ molec$^{-1}$ s$^{-1}$ (Davis et al., 1979).

$$[OH] * t = \frac{-1}{k_{SO_2}} \ln\left(\frac{[SO_2]}{[SO_2]_0}\right) \quad (1)$$

Alternatively, the OH exposure was also estimated using aerosol production from $SO_2$ oxidation. This method assumes complete aerosol conversion of $SO_2$ oxidation. SMPS mass concentrations were calculated using the SMPS size range (9.7 – 422 nm), assuming spherical particles of sulfuric acid density (1.84 g cm$^{-3}$). OH exposure was calculated using Eq. (2) where [SMPS] represents the measured $SO_2$ oxidation aerosol mass concentration (1300 – 4200 µg m$^{-3}$), and [$SO_2$]$_0$

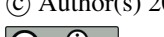


the SO$_2$ gas concentration at the OFR inlet. All aerosol measurements were analyzed for the same 20-min period used for the SO$_2$ gas analysis.

$$[OH] * t = \frac{-1}{k_{SO_2}} \ln\left(1 - \frac{[SMPS]}{[SO_2]_0}\right) \quad (2)$$

### 2.2.2 D$_5$ gas measurement

The reacted D$_5$ concentration was calculated by the difference of upstream and downstream concentration measurements. Duplicate 20-min samples were taken at 0.16 L min$^{-1}$, with flowrate monitored by a mass flowmeter (TSI 4100) calibrated to a bubble flowmeter (Sensidyne Gilian Gilibrator-2). Samples passed through a particle filter (Millipore Millex-FG 0.20 µm filter, CAT SLFG05010) followed by a SPE cartridge (10 mg isolute ENV+ with 1 mL capacity, Biotage AB) (Kierkegaard and McLachlan, 2010; Yucuis et al., 2013). The upstream and downstream gas sampling lines were constantly sampled at
0.16 L min$^{-1}$ to maintain system flows and pressures, regardless of whether active sampling on sorbent was taking place.

### 2.2.3 Method for extraction and instrument analysis of D$_5$ in sample cartridges

During D$_5$ gas phase measurement and analysis, personal care products that contained cyclic siloxanes were avoided by laboratory personnel. Glassware was combusted overnight at 450$^o$C, and then all supplies including glassware were triple rinsed with methanol, acetone, and hexane immediately prior to use. Solvents were pesticide grade from Fisher Scientific.
Cartridges were cleaned by soaking in hexane overnight followed by washing 3 times each with dichloromethane and hexane, respectively. Each cartridge was wrapped in triple-rinsed aluminum foil, sealed in a triple-rinsed amber jar with PTFE-lined screw cap, and kept in a clean media fridge until deployment. Following deployment each sample was re-wrapped in its aluminum foil, returned to its jar, and transferred to the lab for immediate extraction and analysis.

Sample cartridges were eluted with 1.5 mL hexane into gas chromatography (GC) vials. Sample extracts were
spiked with 50 ng PCB 30 (2,4,6-trichlorobiphenyl, Cambridge Isotope Laboratories) immediately prior to instrument analysis as the internal quantification standard. D$_5$ for the calibration standard was purchased from Moravek Biochemicals. Samples were analyzed using an HP 6890 Series GC with an HP 5973 MSD equipped with an Agilent DB-5 column (30 m x 0.25 mm ID, and 1.0 µm film thickness) in select ion monitoring (SIM) mode. Instrument parameters are in Table S2.

Sufficient elution volume was tested by collecting a second cartridge elution of 1.5 mL for the sample with the
highest anticipated concentration, and a cartridge breakthrough test was also performed. Quality Control was assessed through a blank spike test; duplicates; and field, instrument, and method blanks. Values reported herein are not blank corrected. See additional information on these in supplemental information. Non-reactive losses of D$_5$ in the sampling lines and reactor were examined in an experiment without reactor UV illumination. In duplicate testing the sample mass changed by less than 2% between upstream and downstream sampling points.



### 2.2.4 Seed aerosol experiment

Seed aerosol influence on yield was quantified using monodisperse 50 nm ammonium sulfate aerosol (~1000 cm$^{-3}$, 0.2 µg m$^{-3}$) prepared from atomized and dried ammonium sulfate solution (TSI 3076 atomizer, TSI 3062 dryer) with electrical mobility classification. To maintain system pressures between seed and non-seed conditions, seed aerosol entered through a tube with pressure drop equivalent to a HEPA filter, or was filtered through the HEPA filter. The seed influence was tested at a total chamber flow rate of 3.5 L min$^{-1}$, 25% RH, and 80% light intensity. Conditions cycled through $D_5$ only (24-h), $D_5$ + seed (24-h), $D_5$ only (12-h), $D_5$ + seed (12-h), and seed only (12-h).

### 2.3 Hygroscopicity

Hygroscopicity κ parameters were determined by measuring water supersaturation levels required to grow $D_5$ SOA into cloud droplets. A Droplet Measurement Technologies (DMT) cloud condensation nuclei counter CCN-2 (single column; SN 10/05/0024) was used to generate supersaturation conditions and detect activated (grown) particles (Roberts and Nenes, 2005). A CPC (TSI 3785) sampled a side stream before the DMT as a measure of the incoming particles. The controlled variable in the DMT instrument was the thermal gradient (ΔT) which was varied from 0.4 to 24 K through up to eleven steps. Thermal gradients were mapped to supersaturations using monodisperse ammonium sulfate as a calibration aerosol with theoretical supersaturations calculated using the AP3 Kohler model (Rose et al., 2008). Additional details are provided in the supplemental information. The ΔT required for activation of any $D_5$ SOA aerosol was fitted to experimental activation fractions (f; the ratio of DMT and CPC particle counts) using a Gaussian cumulative distribution function.

Eight different monodisperse diameters were tested ranging from 30 to 200 nm. These were produced using a TSI 3080 classifier, TSI long column 3081 DMA, and TSI 3077 neutralizer with sheath air set to 10 L min$^{-1}$. Tubing lengths were the same from the source to the DMT, and the source to the CPC, minimizing the effects of differential particle loss. The DMT instrument was operated with the recommended total flow of ~0.5 L min$^{-1}$ and a 10:1 sheath to sample flow ratio. Ammonium sulfate (Fisher Scientific, purity ≥99%) calibration aerosols were generated using a solution of 1 g L$^{-1}$ prepared in DI water (TSI 3076 atomizer; syringe pump ~15 mL h$^{-1}$; TSI 3062 diffusional dryer) followed by dilution to 2000-3000 cm$^{-3}$. $D_5$ SOA was generated using the OFR with a total chamber flow of 5 L min$^{-1}$, 30% RH, $D_5$ water bath at 70°C, ring flow at 3 L min$^{-1}$, and 100% light intensity. The oxidized $D_5$ aerosol stream was not diluted as concentrations were similar to the diluted ammonium sulfate.

Two issues complicated DMT data analysis. First, temperature and flow spikes were observed that were caused by intermittent faults in a sample temperature sensor. This sensor was part of the instrument feedback control loop used to maintain the column thermal gradient, and accordingly these sensor faults upset the column thermal gradient and required some time to settle. Second, for some of the higher ΔT set points, the temperatures were not able to reach the set points but were stable. To correct for these issues, scripts were developed to automatically classify data into stable (used in data analysis) and unstable (excluded) periods. Data was initially binned to 30 s intervals (raw DMT data was at 1 s, CPC data





was at 5 s). For each 30 s period, temperature, flow, and pressure stability were calculated and compared to thresholds as described below; periods were then flagged as stable or unstable.

Four temperature tests were used: (i) ΔT varied by no more than 0.16 K from the previous 10 s moving average; (ii) T1 (column low temperature) varied by no more than 0.20 K from the previous 10 s moving average; (iii) T3 (column high temperature) varied by no more than 0.20 K from the previous 10 s moving average; and (iv) that the 1 s values of ΔT varied by no more than 0.37 K during the 30 s period. Pressure and flow were checked individually to make sure the relative percent difference between the current value and the 10 s moving average was lower than 4.5%. These data exclusion thresholds were selected by visual inspection of the data, but final data processing was automated. Failure of a single test in any 30 second period led to exclusion from analysis. For periods compliant with these tests (70%), average DMT and CPC concentration were calculated along with the average ΔT.

The Gaussian cumulative distribution function in Eq. (3) was used to fit the measured activation fractions to three parameters (a is the activation threshold, b the activation ΔT, and c the sharpness of the inflection point) which represents the average hygroscopicity, $\kappa_a$ of CCN active particles (Rose et al., 2010). A lower hygroscopicity parameter, $\kappa_t$, was determined from a two parameter (b and c; a=0.5) fit representing the effective hygroscopicity of both active and inactive CCN particles. The two-parameter method is best used when CCN inactive and active particles are externally mixed and for comparison to H-TDMA data (Rose et al. 2010). If the aerosol is of homogenous composition/CCN activity, internally mixed, and there are no counting efficiency errors or particle losses, $\kappa_a$ and $\kappa_t$ should be equal. Deviations in f ≠1 can be used to estimate the fraction of CCN-inactive particles (Rose et al., 2010).

$$f = a\left[1 + erf\left[\frac{(\Delta T - b)}{c\sqrt{2}}\right]\right] \quad (3)$$

The measured diameter and activation supersaturation pairs were used to determine the effective hygroscopicity parameter, $\kappa$ using Eq. (4), the Kappa-Kohler equation. Here s is the supersaturation ratio, $D_{wet}$ is the wet particle diameter which is unknown, D the dry particle diameter taken as the classifier size selected particle diameter, $\sigma_{sol}$ is the surface tension assumed for pure water at the column T1 temperature, $M_w$ is the molecular weight of water, and $\rho_w$ is the density of water at T1. Since the surface tension of pure water is assumed rather than the solution properties (due to the lack of physical properties on the $D_5$ oxidization products) this is the effective hygroscopicity parameter (Rose et al., 2010; Petters and Kreidenweis, 2007). Equation (4) is solved iteratively by guessing an initial $\kappa$, finding the $D_{wet}$ that corresponds to the peak supersaturation from the Kohler curve, and using the $D_{wet}$ term found along with the measured supersaturation to solve for a new $\kappa$ guess which is then substituted into Eq. (4) to find a new $D_{wet}$. Iteration was stopped when $D_{wet}$ and $\kappa$ converged to the experimental supersaturation.

$$s = \frac{D_{wet}^3 - D^3}{D_{wet}^3 - D^3(1 - \kappa)}exp\left(\frac{4\sigma_{sol}M_w}{RT\rho_w D_{wet}}\right) \quad (4)$$



## 2.4 Volatility

D$_5$ SOA was generated as described in 2.3 and then evaporated in a volatility tandem differential mobility analyzer (V-TDMA), previously described in Singh (2015). The V-TDMA featured a 2 s residence time, 1 meter length stainless steel heated and bypass tubes (0.77 cm ID), and a 1 L min$^{-1}$ flowrate. Test aerosols were size selected using electrical classification (TSI 3085) prior to the V-TDMA. Volatility was assessed by comparison of the diameters of unheated and heated aerosols modes, at five temperatures (50, 80, 110, 150, 190°C) and six sizes (10, 20, 30, 50, 80, and 110 nm). These tests were conducted with average aerosol concentrations of 119 – 8x10$^4$ cm$^{-3}$ and 1x10$^{-4}$ – 18 µg m$^{-3}$ (Table S7). In total, 156 samples were collected over 7 h, or two to four replicates at each diameter/temperature combination. Due to some of the very small size changes observed, a continuous number size distribution mode was fit from the discrete SMPS distribution.

## 2.5 Thermal degradation

Vapor phase D$_5$ (estimated mixing ratio of 270 ppm) was generated by bubbling particle free air through liquid D$_5$ using a gas bubbler at 20°C followed by a HEPA filter to remove droplets. The vapor D$_5$ was heated up to 550°C using the V-TDMA system or a stainless steel tube placed in a tube furnace. Both tubes had a residence time of 2 s and the resulting effluent was measured by SMPS to determine if particle formation occurred.

## 3 Results and discussion

### 3.1 OFR chemistry

The OFR chamber conditions tested in this work were evaluated for non-atmospherically relevant chemistry using the OFR Exposure Estimator (v3.1) tool (Lambe and Jimenez). The OFR Exposure Estimator tool was developed using a plug-flow kinetic box model to study the importance of 185 and 254 nm photolysis, O($^1$D), O($^3$P), and O$_3$ chemistry relative to OH (Li et al., 2015; Peng et al., 2016; Peng et al., 2015). Organic radical species have not been assessed in the tool. HO$_2$ radicals are an important component in OFR chemistry, as well as in the atmosphere, but generally are negligible for volatile organic compounds (VOC) oxidation due to low reaction rates relative to OH (Peng et al., 2016). In order to ensure the OFR is run under conditions favoring OH reaction, Peng et al. (2016) recommended an external OH reactivity (OHR$_{ext}$) < 30 s$^{-1}$, H$_2$O mixing ratio > 0.8%, and 185 nm UV flux > 1x10$^{12}$ photons cm$^{-2}$ s$^{-1}$. OHR$_{ext}$ refers to the product of the gas concentration and the OH reaction rate. All experimental conditions used in our OFR experiments were evaluated using the OFR Exposure Estimator tool and fall in the regime with OH chemistry favored. The regime for the personal care product oxidation (Sect. 3.3) could not be determined due to the lack of an estimate of gas concentration. However, the personal care concentrations could be up to 20 times greater than the D$_5$ concentrations used and still fall in the OH-favored regime. Chamber conditions for the 12 scenarios had a water mixing ratio of 0.81 – 1.5% (25-45% RH), OHR$_{ext}$ 0.5 – 26 s$^{-1}$, and 185 nm photon flux as estimated from Li et al. (2015) 6.5 – 8.6 x10$^{13}$ photons cm$^{-2}$ s$^{-1}$.

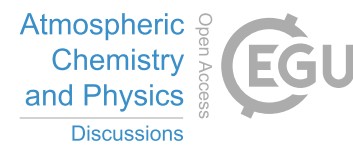

### 3.2 Yield

SMPS aerosol concentrations were corrected to account for particle losses caused by the denuders and tubing downstream of the OFR. Particle loss correction increased the number concentration 7-11%, mass by 3-5%, and decreased the number distribution mode slightly. Upstream and downstream $SO_2$ gas concentrations matched within 9% without OFR illumination, indicating $SO_2$ was not lost to surfaces or leaks. Calculated OH exposure varied from 1.60 to 5.12 x$10^{12}$ molec s cm$^{-3}$, which is equivalent to 12 to 40 days of atmospheric aging assuming an OH concentration of 1.5x$10^6$ molec cm$^{-3}$ (Palm et al., 2016). Measured OH exposures were 4 – 5 times lower than OH exposure predicted from the OFR Exposure Estimator tool in Sect. 3.1. A likely source of uncertainty in the predicted OH exposure is UV light intensity which was estimated from the intensity output from a similar OFR reported in Li et al. (2015). Despite using the same lamp model, light intensity likely varies due to age-related reduction or production differences. Previous evaluation of the predicted OH exposures found values to be within a factor of 3 for 90% of $SO_2$ derived OH estimates (Li et al., 2015).

The $SO_2$ gas phase derived OH exposures were evaluated compared to the estimate derived from $SO_2$ oxidation aerosol measurements. Generally, there was good agreement between the two methods. The aerosol derived estimate of OH exposure, agreed within a factor of 3.6 for experiment 1, and a factor of 1.0-1.8 for the remaining experiments which used higher $SO_2$ concentrations for the $SO_2$ monitor. Generally, the aerosol derived OH estimates were lower than the estimates from $SO_2$ gas reactive consumption. One explanation for lower $SO_2$ derived OH exposure is the potential for loss of sulfate aerosol in the OFR (Lambe et al., 2011a).

$D_5$ gas concentrations (Fig. S4) upstream of the chamber ranged from 290 – 740 µg m$^{-3}$, and downstream from 9 – 24 µg m$^{-3}$. This suggests the reactor consumed nearly all of the precursor gas. Gas measurements had good reproducibility between duplicate samples, and between replicate conditions (trials 2 and 4). Additionally, gas concentrations varied in reasonable ways with respect to dilution flows and to the water bath temperature that was driving $D_5$ evaporation into the system.

A typical yield experiment is shown in Fig. 2. Experiments included a ~20 h settling period for the system to attain a steady state. This was followed by a ~2 h yield determination period, during which the four $D_5$ gas concentrations were measured (2 upstream and 2 downstream) and these were used for determination of reactive organic gas (ΔROG) and associated yield calculation. Similar to Kang et al. (2007), the settling period often had an initial maxima in mass and number during the first few hours. After the yield determination period, $SO_2$ was used to determine OH exposure. Aerosol fluctuations tended to be more stable for the 3.5 L min$^{-1}$ tests than 5 L min$^{-1}$. During the analysis period, aerosol concentrations were reasonably stable in time, with temporal variability (expressed as relative percent difference at 3.5 L min$^{-1}$) of 9 – 11% and 9 – 13% for number and mass concentrations, respectively. At the higher 5 L min$^{-1}$ flow, variability increased to 18 – 26% (number) and 8 – 32% (mass). We suspect variability in the $D_5$ injection rate as a primary cause of variability in the SMPS-detected aerosols from the OFR. Table 1 contains aerosol statistics for the analysis periods.



Yields varied between 0.22 – 0.50 corresponding to aerosol concentrations of 68 – 220 µg m$^{-3}$. Yield was generally invariant of reacted $D_5$, increased monotonically with OH exposure, and generally increased with aerosol mass (Fig. 3). ΔROG did not increase with OH exposure since OH was in excess and $D_5$ was limiting. OH exposure had a strong effect on yield, which is consistent with chemical changes to the products with greater OH exposure. Increasing aerosol yield with

higher OH exposure has been observed in OFR studies with other reactants but yield typically reaches a maximum above which fragmentation reactions likely decrease yield (Lambe et al., 2011a; Lambe et al., 2012; Lambe et al., 2015). No yield maximum with respect to OH exposure was observed. Wu and Johnston (2017) reported aerosol yields of 0.08 – 0.16 at aerosol loadings of 1.2 – 12 µg m$^{-3}$ (at constant OH) with yields increasing with reacted $D_5$ and aerosol loadings. Wu and Johnston (2017) caution that their yields are based on estimated rather than measured $D_5$ consumption.

Wu and Johnston (2017) suggest the aerosol composition is highly dependent on aerosol mass loading with less volatile ring-opened species at low loadings and as particles grow, composition shifts to dimer formation and semi-volatile monomer species. This implies that at the high loadings observed in this work, aerosol composition is likely dominated by semi-volatile monomer species and non-volatile dimer species. In context of available knowledge of aerosol formation for cVMS, Wu and Johnston (2017) observed aerosol size affected aerosol composition, and this work shows the extent of

oxidation may also contribute to aerosol chemistry.

### 3.2.1 Fate of condensable products in the OFR

Condensable vapor (i.e. semi-volatile reaction products) in the OFR can form aerosol, or condense to chamber walls, or undergo fragmentation to non-volatile compounds through multiple OH oxidation, or exit the chamber as a uncondensed semi-volatile gas (Palm et al., 2016). Experiment-specific fates of condensable vapor for the OFR conditions used in this

work were assessed using a condensational loss model (Palm et al., 2016; Lambe and Jimenez). The model compares the relative timescales of competing fates for low volatility organic compounds (LVOC). We modified the LVOC fate model to our chamber conditions for the five yield experiments, and assumed an LVOC diffusion coefficient for the single OH substituted $D_5$ molecule, 4.64x10$^{-6}$ m$^2$ s$^{-1}$ (Janechek et al., 2017). Recommendations for gas diffusion sticking coefficient, OFR volume, eddy diffusion coefficient, number of reactions with OH required to render non-volatile through fractionation,

and surface/volume ratio were from Palm et al. (2016). For the condensational sink calculations (Table S5), the average of the SMPS measurement of the entrance (particle free) and exit was used. For the five yield experiments, the predominant calculated fate is condensation to aerosols (>97%) and aerosol concentrations were not corrected for non-aerosol condensation losses.

Chamber experiments were typically run with particle free incoming air so nucleation must play an important role

in aerosol formation within the chamber owing to the high yields and mass loadings observed. If initial particle growth is fast enough to serve as sufficient condensation sinks for condensable vapors, then the non-aerosol fates are negligible due to the high aerosol concentrations. Alternatively, if the condensational sink is low, the dominant fate is expected to be multi-



generational OH oxidation that could lead to fragmentation. For the range of OH conditions tested in this work, no evidence of fragmentation was observed due to increasing yield with OH exposure.

### 3.2.2 Seed aerosol effect

The seed aerosol influence on aerosol yield was quantified by averaging the final 2 h of SMPS measurements for each of the periods described in Sect. 2.2.4. Here 50 nm ammonium sulfate seed aerosol was added upstream of the reactor to serve as a condensational sink for aerosol formation. The addition of seed aerosols resulted in an increase in number concentration of 2.6%, and mass concentration of 44%. Wu and Johnston (2017) similarly observed positive yield sensitivity to ammonium sulfate seed aerosols.

### 3.3 Personal care product as reactant

Replacing liquid $D_5$ with personal care products as the source of reactive gases similarly generated aerosols when oxidized using the OFR. Using a 10 mg solid flake of antiperspirant as the vapor source led to immediate production of over 2000 µg m$^{-3}$ in the OFR with decay to 600 µg m$^{-3}$ within 1 h, and 60 µg m$^{-3}$ after 4 h. The addition of fresh antiperspirant returned the aerosol concentrations to the initial level. The source of the aerosols was verified to be photooxidation of a personal care product since the cleaned chamber concentrations prior to introduction was 0.03 µg m$^{-3}$, and removing the personal care product or turning off the chamber lights resulted in rapid concentration decrease. The use of 25 mg of conditioner as the vapor source was qualitatively similar but resulted in much lower aerosol loadings of around 40 µg m$^{-3}$, possibly due to lower cVMS content (Wang et al., 2009; Dudzina et al., 2014).

Particle morphology and elemental composition of $D_5$ and antiperspirant generated aerosols were analyzed using electron microscopy. The antiperspirant sample was collected 2 h after the initial flake was added to the system with average SMPS concentrations of 2x10$^6$ cm$^{-3}$, 370 µg m$^{-3}$, and mode of 50 nm. Similar morphology and elemental composition was observed compared to the generated particles using pure $D_5$ (Fig. 4). Particles generated from both sources had spherical morphology with some aggregates. Elemental composition was consistent from a siloxane source with Si and O elemental peaks observed relative to the background. Background peaks of nickel and carbon were observed consistent with the TEM grid.

### 3.4 Hygroscopicity

The CCN measurements up to 1.9% supersaturation were used to calculate the effective hygroscopicity parameter κ for the oxidized $D_5$ aerosols. As described in the methods section, first the DMT temperature differential ΔT was correlated to the supersaturation (S) using theoretical calculations of ammonium sulfate activation supersaturation and measured ammonium sulfate activation ΔT. The result was S = 0.08449 * ΔT - 0.1323 (R$^2$ of 0.994). The linear fit was used to correlate the cVMS 3-parameter and 2-parameter activation ΔT to critical supersaturation needed for κ calculation.



The cVMS measurements (Fig. 5) were corrected for size dependent particle losses and counting efficiencies by dividing by the maximum activation fraction from the ammonium sulfate uncorrected 3-parameter fit. The size dependent maximum activation fractions used were 0.78, 0.87, 0.91, 0.92, 0.94, 0.96, 0.97, and 1.00 for 30, 50, 70, 90, 110, 140, 170, and 200 nm particles, respectively. The remaining deviations from $f_{max} \neq 1$ for cVMS sizes 70-200 nm was attributed to non-homogeneous aerosol composition. Oxidized $D_5$ particles at 30 and 50 nm were too small to exhibit enough activation behavior for calculations (at supersaturations of 1.9%); therefore, $\kappa$ calculations for oxidized $D_5$ started at 70 nm.

The average $D_5$ oxidation aerosol $\kappa_a$ and $\kappa_t$ were 0.011 and 0.0065, respectively. Figure 6 (Table S6) show the $D_5$ oxidation and ammonium sulfate aerosol size dependent effective hygroscopicity parameters as determined by the 3-parameter and 2-parameter fits, labeled $\kappa_a$ and $\kappa_t$, respectively. Typical $\kappa$ values range from 0 for insoluble materials such as soot (Rose et al., 2010) to 1.28 for NaCl (Petters and Kreidenweis, 2007). A large $\kappa$ value corresponds to a species that more easily serve as CCN than a species with a small $\kappa$ value. Zhao et al. (2015) reported examples of secondary organic aerosol $\kappa$ ranging from 0.01 – 0.2, while Lambe et al. (2011b) reported $8 \times 10^{-4}$ – 0.28. The $\kappa$ for oxidized $D_5$ aerosol are on the low end of previously reported SOA. Some literature-reported organic compounds with $\kappa$ below 0.03 include adipic acid and suberic acid (Cerully et al., 2014) and SOA from OFR oxidation of hydrophobic precursors such as C-17 alkane, bis(2-ethylhexyl) sebacate (BES), and engine lubricating oil (Lambe et al., 2011b). The average ammonium sulfate kappa value was calculated using the same algorithm, and a value of 0.79 was obtained. However, there was considerable error for the linear fit correlating DMT $\Delta T$ to supersaturation at low supersaturations for ammonium sulfate particle sizes of 140 – 200 nm (percent error of 30%). This was also observed in Rose et al. (2008) for their low supersaturations.

Figure 6 shows no clear size dependence was observed for $\kappa_t$, while $\kappa_a$ observed smaller particles exhibiting larger kappa values. This suggests the cVMS oxidation aerosol composition was size dependent in our experiments. The observed decreasing trend of $\kappa$ with increasing particle size has been observed before in SOA (Zhao et al., 2015; Winkler et al., 2012) and attributed to smaller particles being more highly oxidized.

## 3.5 Volatility

Particle diameter shift (Fig. S13) between heated and non-heated aerosols indicated the $D_5$ oxidation aerosols produced from the OFR were nearly non-volatile. Particle shrinkage increased with increasing temperature and for smaller sizes which have increased vapor pressure due to particle curvature (Kelvin effect). However, no particles other than 10 nm experienced greater than 4% shrinkage. Particles with 10 nm diameters experienced shrinkage up to 27% but low particle numbers (100-1700 cm$^{-3}$) and temporal variability complicated the interpretation. Number concentrations (Fig. S14) largely stayed unchanged within $\pm$ 15% between heated and unheated cases for particles of size 20 – 110 nm.

## 3.6 Thermal degradation

Since indoor air with high concentrations of cVMS will periodically be contacted against hot surfaces (heating elements of cooking, heating, hobby, and personal care devices), we assessed whether air with $D_5$, heated up to 550°C would result in




aerosol formation. Siloxane and silicone polymer (polydimethylsiloxane; PDMS) are reported as thermally stable (Hall and Patel, 2006; Clarson and Semlyen, 1986), while Erhart et al. (2016) report polymerization of liquid siloxanes at 300° and decomposition at 400°C. We could find no reports in the literature on a thermal degradation temperature for $D_5$ in air, or whether thermal degradation products might tend to condense onto preexisting aerosols or to homogeneously nucleate.

Octamethylcyclotetrasiloxane ($D_4$) has been reported to catalytically polymerize through interaction with a borosilicate container wall when heated to 420°C under vacuum (Clarson and Semlyen, 1986).

No particle formation was observed for any of the tested temperatures (100, 150, 200, 250, and 550°C). With the experimental configuration, condensable decomposition products could occur but elude detection by condensing onto the tubing walls. Therefore, $D_5$ vapor was heated to 550°C and after heating, 80 nm ammonium sulfate seed particles were

introduced to provide a condensational sink for the cooling condensable gases. No growth in the seed aerosols was observed.

### 3.7 Relevance and unanswered questions

In Janechek et al. (2017), the propensity for SOA generation from $D_4$, $D_5$, and $D_6$ across North America was modeled by tracking the OH oxidation products in the gas phase in the CMAQ photochemical grid air quality model. The Janechek et al. (2017) results can therefore be interpreted as the ambient aerosol concentration under the assumption of 100% aerosol mass

fraction (yield). Averaged over the North American modeling domain, the oxidized $D_5$ concentration peaked in summer at ~0.8 ng m$^{-3}$, and the peak monthly averaged concentration occurred downwind of population centers, and was ~10 ng m$^{-3}$. Adjusting these downward using the yields found in this work (0.2-0.5) or to the lower yield measured by Wu and Johnston (2017) (0.08 – 0.16), one would argue that secondary organosilicon aerosols from personal care products should be a very minor mass contributor to outdoor aerosol. However, they should be ubiquitous – present as a widespread and diffuse

background anthropogenic secondary organosilicon aerosol. A possible estimate of the strength of this "signal" is ~0.1 ng m$^{-3}$, based on a reduction of the levels from Janechek et al. (2017) according to the yield experiments of this work.

However, a number of questions remain about this. The assessment done by Janechek et al. (2017) looked only at (a) well-mixed outdoor air (mixed at a spatial scale of 36 km), and (b) $D_4$, $D_5$ and $D_6$ from estimated personal care product use. Other sources of volatile Si were not considered. McDonald et al. (2018) estimated volatile chemical products (VCPs)

represent half the petrochemical derived VOC emissions and the major contributor to potential SOA.

Future questions about secondary organosilicon aerosols include: (a) how relevant are the harsh oxidation (high OH) conditions used in available experimental work for the real atmosphere, (b) what are silicon oxidation product concentrations (gas and particle) in indoor environments and microenvironments such as cars where up to 25 ppb $D_5$ have been observed (Coggon et al., 2018), (c) how well do personal care products with $D_4$-$D_6$ represent all SOA-producing

organosilicon gases, and (d) how do Si-containing and non-silicon containing SOA precursors interact, since both are clearly present from VCPs. While Tang et al. (2015) showed that 31% of molecules offgassed in an indoor environment were cVMS, and McDonald et al. (2018) showed that a large ambient contribution to anthropogenic SOA is from VCPs – the



situation of a substantial anthropogenic organosilicon SOA contribution from Si-containing VCPs cannot yet be ruled out. Reaction of VCPs in the atmosphere can also contribute to $O_3$ formation.

The only probable ambient detection of photochemical organosilicon SOA in the literature is that of Bzdek et al. (2014), who measured a 1-h averaged peak of 5.5 ng m$^{-3}$ of Si using Nano-Aerosol Mass Spectrometer (NAMS) in Pasadena,

California which measures only in the 20-25 nm range. Other detections by the same instrument occurred in coastal Delaware on the US East Coast. Using the fact that cVMS oxidation products are only 38% Si, and that the 20-25 nm range typically represents ~1% of the condensational sink (Bullard et al., 2017), we can extrapolate organosilicon SOA of 1.5 µg m$^{-3}$ from Bzdek et al. (2014). At over 1000 times the expected organosilicon SOA for that location from Janechek et al. (2017), the result raises questions whether the difference is due solely to the reduction in concentration in the model due to

spatial and temporal averaging, or whether there are more fundamental gaps in our understanding of the sources and chemistry of silicon-containing ultrafine and secondary aerosols.

If our understanding of the chemistry, physics, and sources is complete, then issues of spatio-temporal variability can be addressed through finer scale models and comparison to enhanced measurements. For large-scale models that aim to produce a regional, continental, or hemispheric organosilicon signature from cVMS personal care products, the best

available information on yield is shown in Fig. 7, which combines experimental data from this work and Wu and Johnston (2017). Fitting the data to Eq. (5), an Odum two parameter semi-volatile model, results in parameters with yields (α) of 0.14, and 0.82, respectively for saturation concentrations (c*) of 0.95 µg m$^{-3}$ and 484 µg m$^{-3}$. The variable α is the mass based stoichiometric yield, $K_{om}$ the particle phase partitioning coefficient (m$^3$ µg$^{-1}$) which is equivalent to 1/c* (µg m$^{-3}$), $M_o$ the aerosol mass concentration (µg m$^{-3}$), and Y the aerosol yield (aerosol mass fraction). Accordingly, ambient modeling of

cVMS SOA could be performed with these parameters, or with a one product model using the lower volatility species (α of 0.14 at c* of 0.95 µg m$^{-3}$), or a non-volatile product with yield of ~0.1. Information about the temperature dependence of these values is currently unknown.

$$Y = M_o \left( \frac{\alpha_1 K_{om,1}}{1 + K_{om,1} M_o} + \frac{\alpha_2 K_{om,2}}{1 + K_{om,2} M_o} \right) \quad (5)$$

These estimates are subject to the obvious caveats of the high concentration experiments done to date, the uncertainties of applying OFR results to the ambient atmosphere, and the possibility of alternate oxidants or undiscovered aqueous or heterogeneous pathways. But they should be sufficient as a guide for future laboratory, modeling, and field experiments on the subject of organosilicon aerosols.

**4 Conclusions**

This study adds to the short list of laboratory confirmations where aerosol formation from OH oxidation of $D_5$ has been observed and provides one of the first assessments of particle morphology. In this work, a further confirmation of





atmospheric relevance was conducted by verifying that similar aerosols were produced when vapor from solid antiperspirant or from hair conditioner was used as the reactant. A number of important physical properties of these aerosols have now been established, including morphology via electron microscopy, chemical composition (EDS analysis microscopy), sensitivity to seed aerosol, volatility (V-TDMA), and hygroscopicity (CCN activation). We can conclude that in our OFR

experiments $D_5$ with OH produces non-volatile, nearly insoluble aerosols at high yields. Electron microscopy and EDS analysis indicated fractal chain agglomerates were formed in the OFR, with substantial Si and O in the elemental EDS spectra. Using gases emitted from personal care products as the OFR reactant, generated particles with similar morphology and chemical composition compared to the $D_5$ experiments. Yield dependence on OH exposure and size dependence of hygroscopicity suggest that multiple condensable products make up these aerosols. This is consistent with the proposed

mechanism and molecular product assignments of Wu and Johnston (2017).

In a companion paper, aerosols were assessed for acute toxicity in lung tissue during in vitro testing, as well as for propensity to induce biomarkers of inflammation (King et al., 2018). An important caveat of our work is the high concentrations under which these experiments were conducted.

Through integration of this work with other experiments (Wu and Johnston, 2017) and modeling (Janechek et al.,

2017), we have proposed aerosol yield parameterization suitable for atmospheric modeling, and can now estimate the SOA yield from personal care product $D_4$, $D_5$, and $D_6$ cVMS. The SOA from silicon containing personal care products is likely present downwind of many populated areas, and thus may be an additional marker of anthropogenic aerosols. Field confirmation of these values, determination of spatial-temporal patterns, and reconciling of the limited modeling and observational studies are needed.

Additionally, this work provides much needed physical property data and yield parameterization to describe aerosol formation. A recommendation for modelers is to use one of the following, depending on the modeling framework and goals: (a) two parameter semi-volatile model with yield parameters of 0.14, and 0.82, respectively for saturation concentrations (c*) of 0.95 µg m$^{-3}$ and 484 µg m$^{-3}$; (b) one product model using the lower volatility species (α of 0.14 at c* of 0.95 µg m$^{-3}$); (c) a non-volatile product with yield of ~0.1.

**Competing interests**

The authors declare that they have no conflict of interest.

**Acknowledgements**

This research was funded by the National Institute of Environmental Health Sciences through the University of Iowa Environmental Health Sciences Research Center (NIEHS/NIH P30ES005605), the Iowa Superfund Research Program,

National Institute of Environmental Health Sciences (grant P42ES013661), and the National Science Foundation (grant





ATM-0748602). We thank RJ Lee Group for providing the TPS100 sampler and for performing the microscopy and elemental analysis. We also thank Keri Hornbuckle for use of her analytical environmental chemistry lab and for discussions about the research.

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





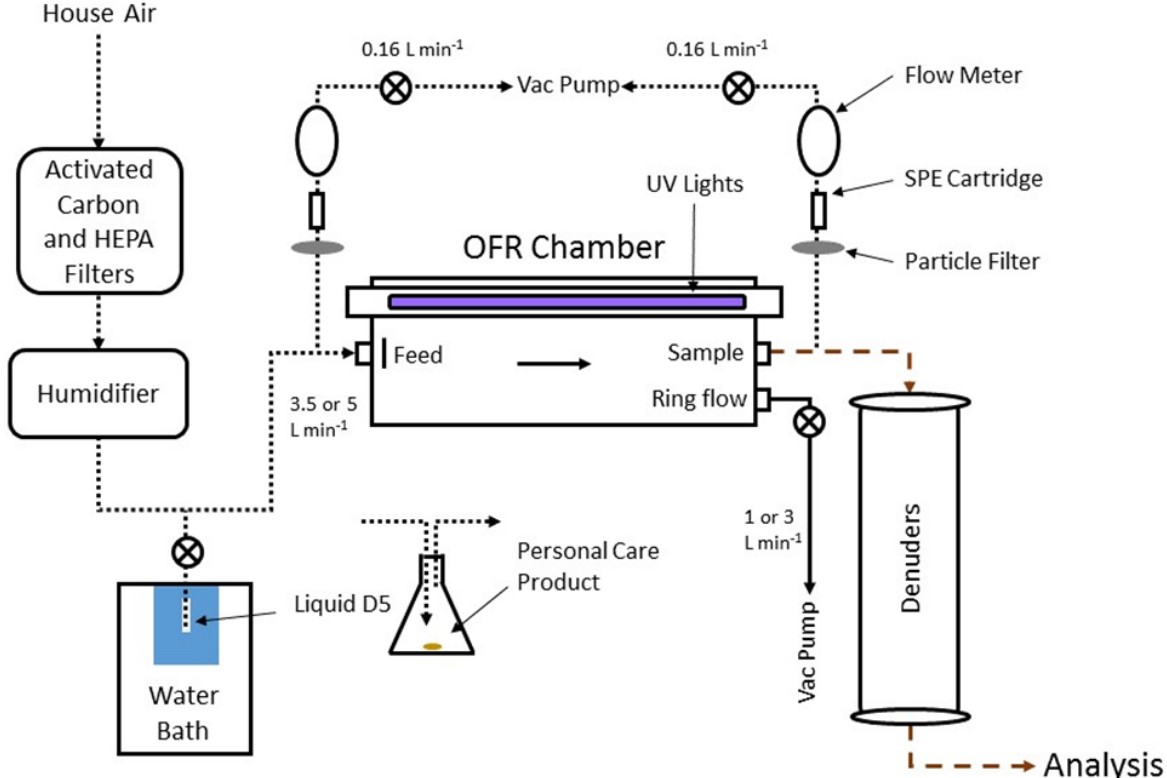

**Figure 1: Flow diagram for generation of aerosols in the OFR. Aerosols were analyzed by SMPS, TPS100, V-TDMA, and DMT-CCN instruments. Delivery of the precursor gas was either by diffusion of liquid D₅ controlled by a water bath or flowing air past a personal care product placed in an Erlenmeyer flask. Short dashed lines in the diagram indicate Teflon tubing, long dashed represent copper tubing, and solid lines represent conductive silicon tubing.**





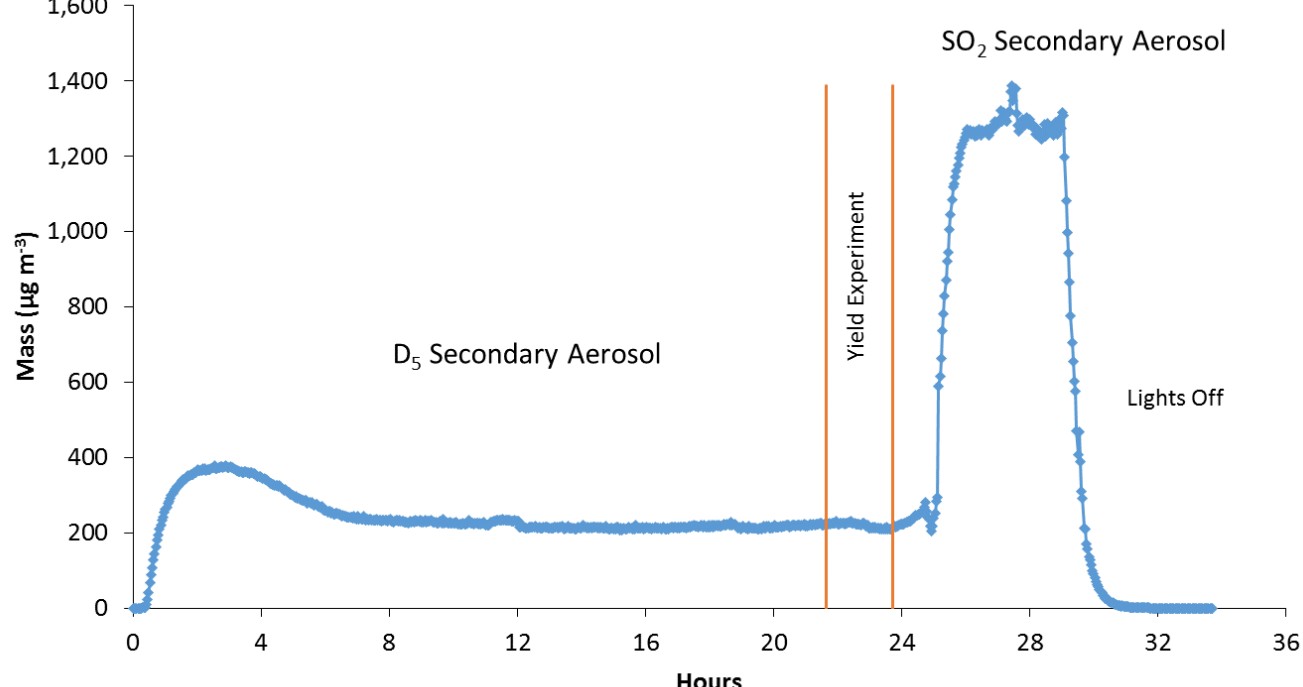

**Figure 2: SMPS time series of a typical yield experiment. An equilibration period was run for ~20 h prior to measuring the D5 gas concentration upstream and downstream of the OFR (yield experiment period; ~1.7 h). Aerosol measurements used for yield analysis were from the yield experiment period. After the yield experiment period, the precursor gas was switched to SO₂ for OH quantification. An SO₂ monitor was used to measure SO₂ downstream of the chamber with and without the OFR lights on.**



**Table 1: Summary of the yield trials. $D_5$ reacted gas concentration represents the average of the upstream minus the average of the downstream measurements. Number, mass, and mode represent the average SMPS aerosol values for the yield experiment period. Equivalent age represents the atmospheric aging assuming an OH concentration of $1.5 \times 10^6$ molec $cm^{-3}$.**

| Trial | Flow (L min$^{-1}$) | RH (%) | Water Bath (°C) | Lights (%) | $D_5$ Reacted (µg m$^{-3}$) | Number (cm$^{-3}$) | Mass (µg m$^{-3}$) | Mode (nm) | OH Exposure (molec s cm$^{-3}$) | Eq. Age (days) | Yield |
|---|---|---|---|---|---|---|---|---|---|---|---|
| 1 | 3.5 | 45 | 70 | 80 | 725 | 3.47E+05 | 219.7 | 83.2 | 4.8E+12 | 37.1 | 0.30 |
| 2 | 3.5 | 25 | 60 | 80 | 356 | 2.57E+05 | 84.0 | 59.4 | 2.3E+12 | 17.4 | 0.24 |
| 3 | 5 | 25 | 70 | 80 | 480 | 3.58E+05 | 107.1 | 69.6 | 1.6E+12 | 12.4 | 0.22 |
| 4 | 3.5 | 45 | 60 | 100 | 358 | 3.07E+05 | 180.7 | 79.8 | 5.1E+12 | 39.5 | 0.50 |
| 5 | 5 | 25 | 60 | 100 | 280 | 3.31E+05 | 68.4 | 57.0 | 2.7E+12 | 20.8 | 0.24 |



**Figure 3: Measured D₅ oxidation aerosol yield as a function of (a) ROG (reacted D₅), (b) equivalent age assuming an OH concentration of 1.5x10⁶ molec cm⁻³, and (c) aerosol mass. Data points are color coded according to OH exposure.**





**Figure 4: STEM-EDS analysis of D₅ oxidation aerosols and antiperspirant oxidation aerosols obtained from analysis of TPS100 samples.**



**Figure 5: D₅ oxidation aerosol CCN activation curve. Size specific κₐ and κₜ are tabulated for particles 70 – 200 nm. The calculated kappa parameter range and average (in parenthesis) is tabulated in the upper left. Each point represents an average 30 s DMT/CPC measurement.**



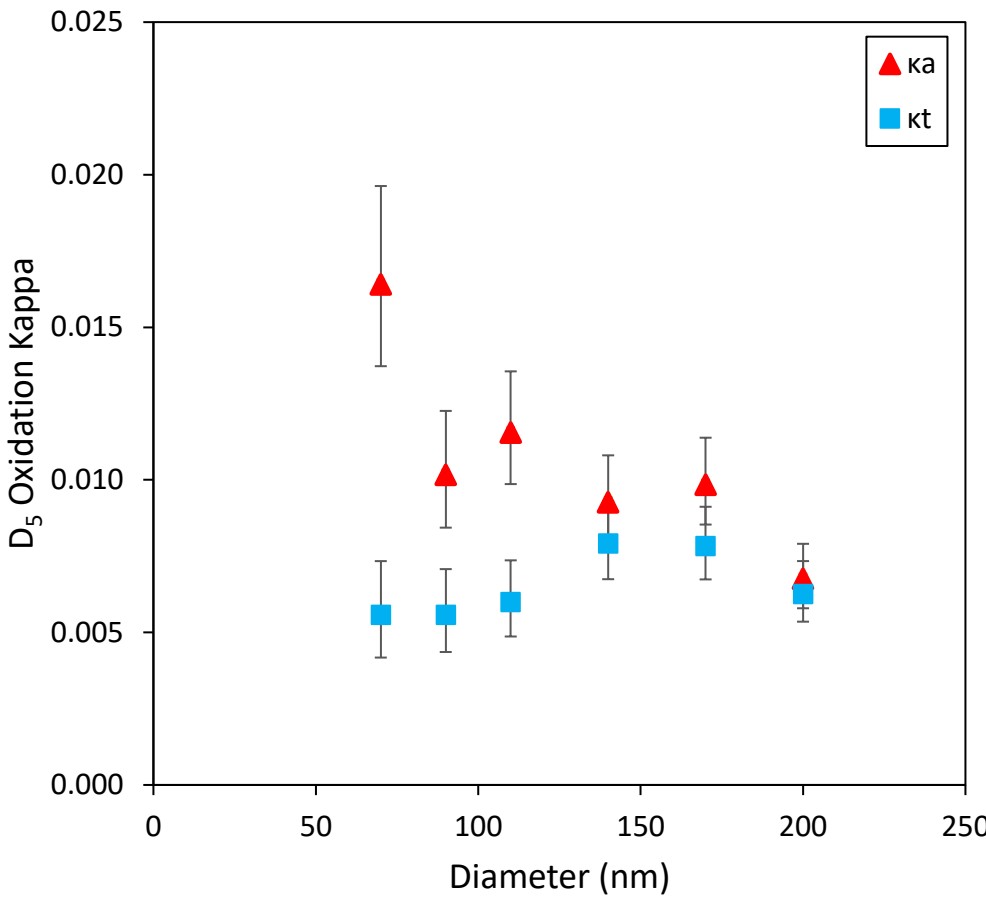

**Figure 6: Size resolved kappa parameters for D₅ oxidation aerosols. Error bars represent the 95% confidence interval.**





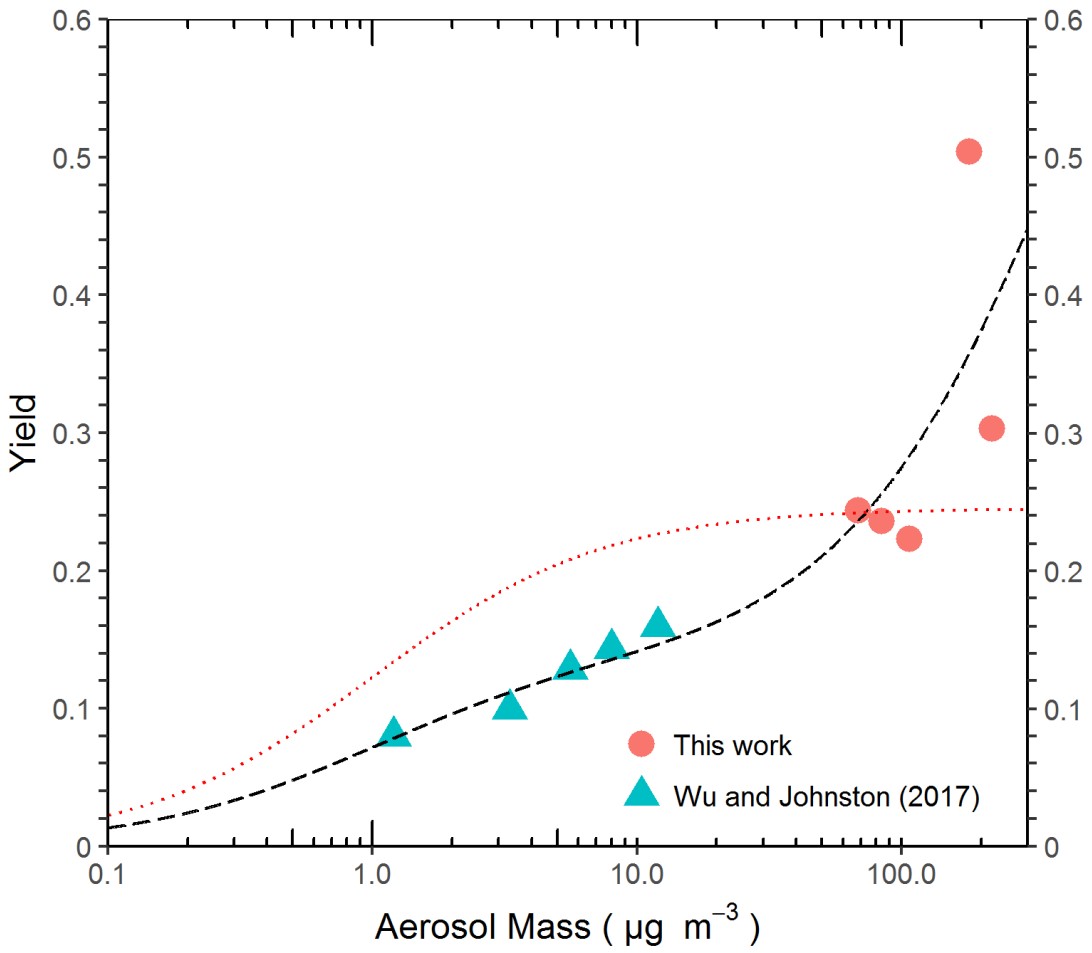

**Figure 7: Two product model fit (black long dash) and one product model fit with saturation concentration (c\*) of 1 µg m⁻³ (red short dash) overlaid with chamber data from this work and D₅ oxidation experiments of Wu and Johnston (2017). Parameters for the two product model fit are yields of 0.14, and 0.82, respectively for c\* of 0.95 µg m⁻³ and 484 µg m⁻³. Parameters for the one product model are a c\* of 1 µg m⁻³ and yield of 0.25.**