# Peer review of "Physical Properties of Secondary Photochemical Aerosol from OH Oxidation of a Cyclic Siloxane"

_Atmospheric Chemistry and Physics, 2018_

## Short Comment (SC1) · 13 Sep 2018

**Emission Fluxes of Siloxanes in the Urban Atmosphere**

Thomas Karl [1], Martin Graus [1], Marcus Striednig [1], Stanislav Juran [2]

[1]Institute for Atmospheric and Cryospheric Sciences, University of Innsbruck, Innsbruck, Austria

[2] Global Change Research Institute, Czech Academy of Sciences, Brno, Czech Republic.

The paper by *Janechek et al.* investigates the atmospheric fate of cyclic volatile methyl

siloxanes (cVMS) in laboratory (ie. flow reactor) experiments, and highlights the need of a better representation of these compounds in air quality models. While there is already an impressive body of literature on indoor and outdoor concentration measurements of siloxanes, it may not be widely recognized, that recently, urban flux data for D3-D6 siloxanes are also available, allowing to estimate area weighed emissions into the urban atmosphere. Based on these direct eddy flux measurements, urban emission patterns of siloxanes can be clearly separated from conventional combustion NMVOC (e.g. aromatics such as benzene; *Karl et al., 2018*). Based on PTR-qlTOFMS eddy covariance measurements we have also observed different emission patterns among various siloxanes suggesting multiple urban sources for these species, in line with the wide application range of cVMS. New data in Innsbruck for the summer of 2018 show typical daytime area weighted emission fluxes of D3+D4+D5+D6 siloxanes around  6 pmol/m2/s (Fig 1 panel A). Siloxane mixing ratios are typically highest during night (up to approx. 6 pptv), when smaller but persisting emissions accumulate in the shallow nocturnal boundary layer (Fig. 1 (B)). Compared to typical combustion related NMVOC (e.g. benzene: Fig 1 C and D), siloxane emission fluxes tend to be more persistent during night, and particularly more pronounced during the weekday rush hour (Fig 2.). While the cyclic D5 siloxane dominates the atmospheric siloxane flux (approx. 50 %), in line with earlier findings, the contributions from D3 and D4 siloxanes (Fig 3) emitted into the urban atmosphere are also significant and seem higher compared to some reported indoor distributions (e.g. *Tang et al., 2015*), which highlights the need to quantify actual emissions into the atmosphere.

*Siloxanes* were detected on the following mass to charge ratios using PTR: D3 ('m223.0637 $(C_6H_{18}O_3Si_3)H^+$'), D4 ('m297.0824 $(C_8H_{24}O_4Si_4)H^+$'), D5 ('m355.0699 $(C_9H_{27}O_5Si_5)^+$' (fragment) and 'm371.1012 $(C_{10}H_{30}O_5Si_5)H^+$' (parent)), D6 ('m445.1200 $(C_{12}H_{36}O_6Si_6)H^+$')

**References:** Karl T., et al. Urban flux measurements reveal a large pool of oxygenated volatile organic compound emissions, PNAS, 115, 1186-1191.

https://doi.org/10.1073/pnas.1714715115, 2018.

Tang, X., et al.: Siloxanes are the most abundant VOC emitted from engineering students in a classroom, Environmental Science  Technology Letters, DOI: 10.1021/acs.estlett.5b00256, 2015.

[Figure]

[Figure]

**Fig. 1.** Diurnal cycles of cVMS and benzene fluxes: (A) the sum of D3, D4, D5 and D6 flux; (B) ambient mixing ratios; (C) the mass flux ratio of siloxanes over benzene, and (D) benzene fluxes.

[Figure]

**Fig. 2.** Median diurnal flux profiles of siloxanes for weekdays (i.e. Tuesday-Thursday) and Sunday.

[Figure]

**Fig. 3.** Relative distribution of molar siloxane fluxes

---

## Referee Comment (RC1) · Anonymous Referee #2 · 1 Oct 2018

Janechek et al. provide new laboratory measurements of SOA formed from the OH oxidation of cyclic volatile methyl siloxanes. The authors produce SOA via an OFR and test various aerosol properties, including morphology, composition, volatility, and hygroscopicity. The authors also explore various experimental parameters on SOA yield, and report new yields that are higher than previously measured.

The results from this study are useful for models aimed at understanding organosilicon aerosol properties. I'm particularly interested in the TEM measurements and how these could be useful for identifying organosilicon aerosols in the atmosphere. Overall, I find the methods to test aerosol properties to be good and the conclusions about volatility, hygroscopicity, and composition to be sound. I have some concerns about the aerosol yield conclusions (see below). I would appreciate if the authors addressed these concerns prior to publication.

**Major Comments:**

Page 11, lines 1 -15 and Page 11, lines 29-32.

I have some concerns with the interpretation of the yield experiments. How certain are the authors that the yield measured at the highest OH exposure is statistically significant? Looking at panels A and C, it seems like this point could be an outlier, but the authors attribute the uptick in yield as an increase in semi- or low-volatility oxidation products. The authors argue that the fate of gas-phase species is condensation to aerosols (line 27), but then note that the experiments were run in particle-free air. I interpret this to mean that the authors are relying on nucleation processes to drive particle formation. If so, this significantly complicates the interpretation of the yield because there is no way of understanding the nucleation rate and its relationship with the vapor condensational sink. Consequently, I believe that the authors are overestimating the aerosol sink and underestimating the aerosol yield. The authors even show this experimentally in the following section (3.2.2). When the authors add 50 nm ammonium sulfate particles, the aerosol mass increases by 44%. Would this not suggest that the maximum yield was not observed for any of the experiments conducted with particle-free air, and that significant condensational losses were to the walls of the chamber?

In yield experiments, it is crucial that sufficient aerosol be added in order to drive oxidation products away from the walls and to the aerosol phase (Zhang et al. 2014). Consequently, I believe that the seed aerosol experiment described in section 2.2.4 should actually be treated as the yield experiment. If this input of aerosol is significant enough drive most of the oxidation products to the aerosol phase, then I believe the authors can interpret the effect of OH exposure on aerosol yield; otherwise, I would treat this yield as a lower-limit estimation.

Section 3.3

I find this section to be quite interesting, though the discussion is very brief. I understand that the primary purpose of this section is to show that the aerosol collected from antiperspirant oxidation exhibits a qualitatively similar TEM spectra as aerosol produced from D5 oxidation. Does this also suggest that D5 was the predominant SOA precursor in the antiperspirant experiment and that other components of the antiperspirant did not contribute to SOA formation? I presume that there were a lot of other components in the antiperspirant that was tested, including fragrances (which were likely a mixture of highly reactive compounds like monoterpenes and terpenoids) and possibly other ingredients, such as glycols. Can the authors provide more details of the ingredient list so that the reader has a sense of what precursors may have been present in the antiperspirant stick? Was D5 listed at the top of this list, which presumably implies that it was a major ingredient?

Did the authors also test the composition of the aerosols formed via oxidation of conditioner emissions? I'm curious if the TEM measurements show the same relative elemental composition, or if other peaks (such as carbon) are more abundant. If this spectra is different, then this could point towards the influence of other ingredients on SOA formation.

Ultimately, I think the authors can do a bit more here to put the personal care product experiments into perspective. As written, a reader could interpret these results to suggest that aerosol formed from personal care products containing D5 is exclusively composed of D5 oxidation products! This is unlikely, of course, and not what the authors are intending to show, so some further discussion should be provided. Furthermore, could the TEM technique be useful in understanding organosilicon aerosol in the atmosphere? Presumably, most particles containing silicon in modes < 100 nm would have been derived from secondary processes. Could these TEM measurements be a useful tool in identifying aerosols resulting from organosilicon oxidation?

Page 15, lines 3-11.

In this section, the authors discuss the relevance of cVMS SOA formation and potential shortcomings of this work. The authors rightfully demonstrate that cVMS will contribute a small fraction to total SOA; however, I believe the authors are overextending when they try to frame these results into the measurements conducted by Bzdek et al. (2014). Specifically, I take issue with the extrapolation of the organosilicon SOA content. The nano-particle measurements conducted by Bzdek et al. provide a constraint to nucleation processes close to the source, but downwind of major cities like L.A., chemical and meteorological conditions could change and limit the oxidation of organosilicon precursors. Furthermore, an extrapolation to 1.5 ug/m3 seems quite unrealistic. That amount of SOA is comparable to the background SOA in LA believed to come from regional biogenic sources (~2 ug/m3, Hayes et al. 2015).

I recommend that the authors refrain from extrapolating to suggest that there could be unexplained sources of organosilicon precursors. I think it is sufficient to cite Tang et al. (2015) and McDonald et al. (2018) to show that organosilicon SOA precursors are emerging as an

important source of VOCs in urban areas, and that incorporating this work into models will help to constrain the VCP impact on urban SOA formation.

**Other Comments**

Page 2, Lines 8 - 10. This sentence seems a bit out of place here, and doesn't necessarily requires its own paragraph. I suggest moving it elsewhere in the introduction ( perhaps after the last sentence at line 32?)

Page 3, Lines 1-20. This material would be better presented as a discussion rather than as an introduction. I would recommend moving this to section 3.1

Page 3, Line23-24. This sentence reads as if concentration, size, and morphology were also measured by EDS. Please rephrase.

Page 4, Line 6. Please add "by photolysis of water" after "in situ"

Page 4, Line 11. It would be clearer to say "temperature-controlled"

Page 4, Line 14. It would be clearer to the reader if you mention that cyclomethicone and cyclopentasiloxane often refer to the same molecule.

Page 5, Line 5. Please note that the "D5 water bath temperature" translates to variations in precursor concentrations.

Page 5, Lines 11 - 12. From my understanding, it's not crucial that the concentrations be the same; rather, it's important that the OH reactivity be low enough that losses due to OH titration can be ignored. What concentration range of D5 and SO2 was being injected into the OFR?

Page 5, Line 21. The term "disappearance" suggests that the SO2 was lost by some unknown process. I would recommend replacing with "…measurement of reacted SO2 in the OFR".

Page 5, Line 25. Please add "constant" after "rate".

Page 6, line 2. How well do the OH exposures calculated from Eqns 1 and 2 agree?

Page 6, Line 26. The semi-colons should be replaced with commas.

Page 7, Line 12. I would refer to "the DMT" or the "DMT Instrument" as a "CCN counter"

Page 7, Line 12-13. Consider removing the phrase "The controlled variable in the DMT instrument" and replace with "The thermal gradient was varied from…" for brevity.

Page 7, line 20. By "source", do you mean from the DMA monodisperse outlet?

Page 7, line 27 - Page 8, Line 10. The details of data QA are not needed here. I suggest removing, or placing in the supplemental information.

Section 2.5. I'm somewhat confused by the thermal degradation experiments. The authors don't explain in detail why these experiments were performed, or why these might be relevant in the atmosphere. Are there cases in which D5 might be exposed to high temperature that it would decompose and form lower volatility components? Or, is this a check to evaluate potential biases in the volatility experiments? If it is the latter, I would suggest moving this discussion to Section 2.4 and refer to these experiments as controls.

Page 10, line 21. What do the authors mean by "reasonable ways?"

Page 10, line 23. It seems to me that the experiment reached steady state long before 20 hrs! No need to change - I'm just impressed by the length of the experiment.

Page 10, line 24 - 25. This sentence is a bit akward. I might consider rephrasing as "This was followed by a sampling period, during which four gas samples were taken to determine D5 SOA yields."

Page 10, line 31. I'm confused why the authors implicate the D5 injection rate as the primary cause for variability. Is it a mixing issue?

Page 13, line 19. Please add "that" between "shows" and "no". Also, it is incorrect to write "ka *observed* smaller particles" - please choose a different verb.

Page 13, lines 20-22. Can the authors please clarify the last statement about the higher oxidation state of smaller particles? Is that because the larger particles contain more semi-volatile components (e.g. dimers, or second-generation monomers?).

Section 3.5, Figure S13. The 10 nm mode is quite noisy. If it is difficult to quantify, why include this mode in the figure? I would find this figure more useful if I could see the zoomed-in traces for 20-110 nm experiments.

Section 3.6. Would thermal degradation be a significant process influencing the lifetime of D5? If so, please provide some justification. If not, I feel like these experiments are better characterized as volatility control experiments - i.e., to ensure that changes in particle concentrations during volatility experiments are not the result of residual gaseous D5 that could degrade and lead to particle formation.

**References**

Zhang et al. (2014). Secondary organic aerosol formation in chambers. Proceedings of the National Academy of Sciences, 111 (16) 5802-5807

Hayes et al. (2015). Modeling the formation and aging of secondary organic aerosols in Los Angeles during CalNex 2010, Atmos. Chem. Phys., 15, 5773-5801

---

## Referee Comment (RC2) · Anonymous Referee #1 · 7 Nov 2018

Janechek et al. describe yields and physical properties of SOA formed from the photooxidation of the D5 volatile siloxane and D5-containing consumer product. Such studies are definitely important because we know very little about the atmospheric fates of these cyclic volatile methylsiloxanes (cVMS) and the study could be useful for an initial step in more accurate modelling which currently is also challenging. A rather shocking discovery from this study is that these volatile silicon-containing organic compounds appear to have much higher yields for secondary organic aerosol (SOA) formation than previously thought which might inspire new questions about how SOA from the large anthropogenic emissions of siloxanes affect human health and climate.

[Figure]

Overall, the paper is well written, and the atmospheric relevance of siloxanes is a timely subject, so it looks like a useful contribution to the literature. However, I have some relatively minor suggestions/comments which hopefully can be successfully addressed during the revision.

**General**

- Why did the authors focus only on D5 experiments? While this is indeed the most common cVMS in consumer care products, other siloxanes are also common in the atmosphere (as the nice community comment emphasizes). I think the authors should consider extending the analysis to other cyclic siloxanes (D3-D7) but if it is not possible then at least the other compounds should be discussed in terms of how similar or different they might be. I am particularly concerned that your models extrapolate from this study to D4 and D6 so might be completely inaccurate for the mixed siloxane atmospheres (D3-D7).

**Specific**

- The methodology is very nicely described. However, given the high SOA yields, are you sure that the house air was not introducing any additional precursors or cVMS? Why was the house air used to flow over D5 standard instead of zero air from a catalyst or a zero-air generator? The indoor air may contain very high concentrations of siloxanes and other VOCs (e.g. Tang et al., 2016). It is quite reassuring what is written in P6 L12 "During D5 gas phase measurement and analysis, personal care products that contained cyclic siloxanes were avoided by laboratory personnel." but was the personnel instructed to use no shampoo, soap, creams, or just to avoid antiperspirants? There could potentially be grease or products which could be a strong siloxane source in a lab. I therefore wonder how the authors have convinced themselves there was no significant level of VOCs (e.g. 100 ppb of D4, for instance)? Your assumption of the clean air might be due to the presence of a charcoal filter but

was it new and how efficient was it for cVMS? Have you done any measurements of the air before and after the charcoal filter before each yield experiment? I think a zero-air generator would be a much better solution for these type of experiments.

- Brass fittings are avoided in VOC sampling. Is there any reason why these were used?

- Fig. 3, is one of the points an outlier? Fig S8 informs that these yield extremes occur at the RH of 45%, and are more consistent at RH of 25%. It would be nice to shed more light on understanding the effect of humidity.

- Table 1, water bath temperature only affected the evaporation rate of D5 to the dilution flow. It would be useful to add temperature of the reactor.

**Technical**
- P7,L32 Should be "Data were".

**References**
Tang X, Misztal PK, Nazaroff WW, Goldstein AH. Volatile Organic Compound Emissions from Humans Indoors. Environmental Science Technology, 50(23):12686-94, 2016.

---

## Author Comment (AC1) · 20 Dec 2018

Please see the corresponding supplement to this comment for our response to the two reviewers and interactive comment from Karl et al.

Please also note the supplement to this comment:
https://www.atmos-chem-phys-discuss.net/acp-2018-748/acp-2018-748-AC1-supplement.pdf

---

## Author Response (AR1)

**Response to review comments regarding ACPD Janechek et al. "Physical Properties of Secondary Photochemical Aerosol from OH Oxidation of a Cyclic Siloxane"**
**https://www.atmos-chem-phys-discuss.net/acp-2018-748/**

**Response to Interactive Comment:**

*The coauthors thank the two anonymous reviewers for their comments, and the interactive comment from Karl et al. (Emission Fluxes of Siloxanes in the Urban Atmosphere; Thomas Karl, Martin Graus, Marcus Striednig, Stanislav Juran; Institute for Atmospheric and Cryospheric Sciences, University of Innsbruck, Innsbruck, Austria; and Global Change Research Institute, Czech Academy of Sciences, Brno, Czech Republic).*

The interactive comment from Karl et al. can be found at https://doi.org/10.5194/acp-2018-748-SC1

> *We appreciate the comment from Karl et al. regarding improved information for direct measurement of cVMS fluxes. We have updated the paper's introduction to mention and cite Karl et al. (2018).*
>
> *Added mention of flux measurements to page 2, line 5-6, and citation to page 2, line 26.*

**Response to Anonymous Referee #1:**

Reviewer comment:

Janechek et al. describe yields and physical properties of SOA formed from the photooxidation of the D5 volatile siloxane and D5-containing consumer product. Such studies are definitely important because we know very little about the atmospheric fates of these cyclic volatile methylsiloxanes (cVMS) and the study could be useful for an initial step in more accurate modelling which currently is also challenging. A rather shocking discovery from this study is that these volatile silicon-containing organic compounds appear to have much higher yields for secondary organic aerosol (SOA) formation than previously thought which might inspire new questions about how SOA from the large anthropogenic emissions of siloxanes affect human health and climate.

Overall, the paper is well written, and the atmospheric relevance of siloxanes is a timely subject, so it looks like a useful contribution to the literature. However, I have some relatively minor suggestions/comments which hopefully can be successfully addressed during the revision.

> *The co-authors appreciate the supportive comment from reviewer 1.*

1. Why did the authors focus only on D5 experiments? While this is indeed the most common cVMS in consumer care products, other siloxanes are also common in the atmosphere (as the nice community comment emphasizes). I think the authors should consider extending the analysis to other cyclic siloxanes (D3-D7) but if it is not possible

then at least the other compounds should be discussed in terms of how similar or different they might be. I am particularly concerned that your models extrapolate from this study to D4 and D6 so might be completely inaccurate for the mixed siloxane atmospheres (D3-D7).

*$D_5$ was chosen because it is the most common cyclic siloxane in personal care products. It is also often the highest concentration cVMS measured in studies that look at multiple cVMS compounds. Extending the work to $D_3$, $D_4$ or $D_6$ or to linear siloxanes is an excellent idea, but would need to be future work. We added mention in the paper (p. 17, lines 21-24) that extension to other siloxanes is uncertain due to the lack of yield measurements and extension to other species, based on volatility of the precursors (volatility: $D_4 > D_5 > D_6$ (Lei et al., 2010).*

2. The methodology is very nicely described. However, given the high SOA yields, are you sure that the house air was not introducing any additional precursors or cVMS? Why was the house air used to flow over D5 standard instead of zero air from a catalyst or a zero-air generator? The indoor air may contain very high concentrations of siloxanes and other VOCs (e.g. Tang et al., 2016). It is quite reassuring what is written in P6 L12 "During D5 gas phase measurement and analysis, personal care products that contained cyclic siloxanes were avoided by laboratory personnel." but was the personnel instructed to use no shampoo, soap, creams, or just to avoid antiperspirants? There could potentially be grease or products which could be a strong siloxane source in a lab. I therefore wonder how the authors have convinced themselves there was no significant level of VOCs (e.g. 100 ppb of D4, for instance)? Your assumption of the clean air might be due to the presence of a charcoal filter but was it new and how efficient was it for cVMS? Have you done any measurements of the air before and after the charcoal filter before each yield experiment? I think a zero-air generator would be a much better solution for these type of experiments.

*In hindsight, the use of air from the zero air generator would be preferable. However, we checked for SOA formation from the purified air (without addition of $D_5$). Finding negligible concentrations, we felt confident moving forward with our combination of "zero" air and filtration/purification of the house air. We felt this was sufficient quality assurance that we were correctly attributing yield to the $D_5$. We do not have a measurement of the siloxane levels before or after the charcoal filter, as we felt our quality assurance was sufficient. After cleaning the chamber (15 hours) background aerosol concentrations were 400 $cm^{-3}$ and 0.001 $\mu g\ m^{-3}$ (with the OFR lights on and RH addition). This additional information has been added to the manuscript on page 4, lines 23-25.*

*To clarify the avoidance of personal care products, personnel directly handling the sorbent cartridges (and then doing extraction, analysis, handling blanks, etc.) avoided products (all personal care products) with cyclic siloxanes on the product labels. Gloves were used to handle all samples. The supplement contained extensive information on field blanks, method blanks, and duplicate sample. These indicate that precision and accuracy*

*of our D₅ sampling was acceptable and that procedures to avoid contamination were sufficient.*

3. Brass fittings are avoided in VOC sampling. Is there any reason why these were used?

   *In hindsight we should have used Teflon fittings rather than brass in conjunction with the Teflon tubing used. While future work should use Teflon fittings, we argue the impact of the brass fittings should be minimal.*

   *We only quantified D₅ during this work, used high concentrations, long equilibration times, and minimal surface area of metal fittings. Recent research on the interaction of VOCs with tubing have been investigated for a range of compounds and materials but not brass (Pagonis et al., 2018; Pagonis et al., 2017). Metal tubing has shown increased VOC interactions compared to Teflon due to adsorption. However, we believe based on these studies, VOC interaction with metals for the conditions used in this work would be negligible based on several reasons:  1) VOC adsorption has been found to be small even at low concentrations for compounds with volatility comparable to D₅ (saturation concentration of $3 \times 10^6$ µg m$^{-3}$ (Lei et al., 2010)), 2) vapor concentrations used in this work were in excess of concentrations that showed quick saturation of tubing adsorption sites, 3) tubing adsorption is reduced in the presence of humidified air greater than 20% RH needed for OFR OH production, and 4) the surface area and contact time between gases and brass was small (Coggon, 2018; Jimenez, 2018).*

4. Fig. 3, is one of the points an outlier? Fig S8 informs that these yield extremes occur at the RH of 45%, and are more consistent at RH of 25%. It would be nice to shed more light on understanding the effect of humidity.

   *We are not sure if it is an outlier or not. That trial did have a difference from the other trials (see Table 1); it was the only one with 100% power on lights, and high (45%) RH. Since we have no good reason to exclude it, we feel it is best to leave it in the dataset. Our assumption is that readers will see the imprecision in the yield results, and take that as a call for future research to establish more certain and more precise yields – with varying OH concentrations. In other words, we used the OFR primarily as an aerosol generation system to generate the aerosols and subject them to physical tests (such as hygroscopicity). While we were doing that, it was appropriate to also investigate yield. But readers will see (without us having to call extra attention) that the yield results are in need of future studies for confirmation, investigation of chemical mechanism and molecular composition, investigation of artifacts that may be caused by the OFR or other methodological details. Although we already had several caveats in the article, we have added more to stress the yields need replication. We have changed the last sentence of the abstract (p. 1, line 23-24); we have added a caveat about replication into the conclusions (p. 18, lines 10-12). And we have called out the potential outlier in results and discussion (p. 12, lines 20-25).*

5. Table 1, water bath temperature only affected the evaporation rate of D5 to the dilution flow. It would be useful to add temperature of the reactor.

*We did not measure the internal OFR temperature but rather the incoming air flow which was held constant at room temperature of 22 °C. This is also consistent with reported temperatures from SMPS data files that were measured downstream of the chamber. However the actual OFR temperature was likely slightly warmer due to the lamps. Li et al. (2015) reports that this OFR heating is likely ~2 °C. We have revised the manuscript (p. 4, lines 14-16) to add the estimated reactor temperature and state that no temperature correction has been performed on measurements to account for the reactor being slightly warmer.*

6. P7,L32 Should be "Data were".

   *Thanks for the correction. We have fixed the verb.*

**Anonymous Referee #2:**

Janechek et al. provide new laboratory measurements of SOA formed from the OH oxidation of cyclic volatile methyl siloxanes. The authors produce SOA via an OFR and test various aerosol properties, including morphology, composition, volatility, and hygroscopicity. The authors also explore various experimental parameters on SOA yield, and report new yields that are higher than previously measured.

The results from this study are useful for models aimed at understanding organosilicon aerosol properties. I'm particularly interested in the TEM measurements and how these could be useful for identifying organosilicon aerosols in the atmosphere. Overall, I find the methods to test aerosol properties to be good and the conclusions about volatility, hygroscopicity, and composition to be sound. I have some concerns about the aerosol yield conclusions (see below). I would appreciate if the authors addressed these concerns prior to publication.

1. Page 11, lines 1 -15 and Page 11, lines 29-32. I have some concerns with the interpretation of the yield experiments. How certain are the authors that the yield measured at the highest OH exposure is statistically significant? Looking at panels A and C, it seems like this point could be an outlier, but the authors attribute the uptick in yield as an increase in semi- or low-volatility oxidation products.

   *This is a very similar comment to reviewer 1, comment 4. We don't have specific information that it is an outlier, and the test conditions are unique. Therefore, the appropriate action (in our opinion) is to report it into the literature and then let it be revisited by future work. As we responded above (reviewer 1, comment 4) we have made the caveats to our yields more consistent throughout the paper.*

2. The authors argue that the fate of gas-phase species is condensation to aerosols (line 27), but then note that the experiments were run in particle-free air. I interpret this to mean that the authors are relying on nucleation processes to drive particle formation. If so, this significantly complicates the interpretation of the yield because there is no way of understanding the nucleation rate and its relationship with the vapor condensational sink. Consequently, I believe that the authors are overestimating the aerosol sink and underestimating the aerosol yield. The authors even show this experimentally in the following section (3.2.2). When the authors add 50 nm ammonium sulfate particles, the aerosol mass increases by 44%. Would this not suggest that the maximum yield was not observed for any of the experiments conducted with particle-free air, and that significant condensational losses were to the walls of the chamber?

   In yield experiments, it is crucial that sufficient aerosol be added in order to drive oxidation products away from the walls and to the aerosol phase (Zhang et al. 2014). Consequently, I believe that the seed aerosol experiment described in section 2.2.4 should actually be treated as the yield experiment. If this input of aerosol is significant enough drive most of the oxidation products to the aerosol phase, then I believe the authors can

interpret the effect of OH exposure on aerosol yield; otherwise, I would treat this yield as a lower-limit estimation.

*In the paper, the loss to the walls was estimated using a condensational loss model (Palm et al., 2016; Lambe and Jimenez, 2018) which indicated that loss to the walls (and other non-aerosol fates) was minimal, under the assumption that the average of the condensational sink at the entrance (assumed to be zero) and the condensational sink at the exit (measured) was representative of the whole OFR. In other words, the condensational sink assumed was one half that measured at the exit. Although not reported in the paper or the supplement, we have run the condensational loss model for a number of different assumptions for condensational sink. Specifically, we investigated one order of magnitude lower, and three orders of magnitude lower. Only in the case of the condensational sink being 3 orders of magnitude lower did the calculator estimate that condensation to aerosol was not the dominant pathway and even then the dominant pathway was reacting with OH more than 5 times. However, after considering the reviewers comment, Zhang et al. (2014), and the point regarding the yield increase with seed aerosol, we agree that reporting the yields as a lower limit is appropriate. We have changed text in the abstract (p. 1, lines 20-21), results and discussion (p. 13, lines 19-22), and conclusions (p. 18, lines 20-21) sections to reflect this.*

3. Section 3.3. I find this section to be quite interesting, though the discussion is very brief. I understand that the primary purpose of this section is to show that the aerosol collected from antiperspirant oxidation exhibits a qualitatively similar TEM spectra as aerosol produced from D5 oxidation. Does this also suggest that D5 was the predominant SOA precursor in the antiperspirant experiment and that other components of the antiperspirant did not contribute to SOA formation?

   *That is our working hypothesis but we only have limited proof, from the similarity of the SEM-EDS microscopy done by our collaborator RJ Lee. Morphology, size distribution, and EDS spectra were similar between the two sources.*

4. I presume that there were a lot of other components in the antiperspirant that was tested, including fragrances (which were likely a mixture of highly reactive compounds like monoterpenes and terpenoids) and possibly other ingredients, such as glycols. Can the authors provide more details of the ingredient list so that the reader has a sense of what precursors may have been present in the antiperspirant stick? Was D5 listed at the top of this list, which presumably implies that it was a major ingredient?

   *Dudzina et al. (2014) reported that the median $D_5$ mass fraction in antiperspirant was 0.142. This, combined with the elemental spectra of the SOA in EDS (with a prominent Si peak), leads us to believe that a cyclic siloxane (presumably $D_5$) was the major constituent being oxidized. While we don't know the quantitative breakdown, $D_5$ was listed as the first inactive ingredient on the product label implying it is a major ingredient. During the Access Review for ACPD, the editor and coauthors agreed that the specific manufacturer and product name would not be released with the article, as the focus on the article was on the general class of personal care products, not specific*

*products. Thus we are refraining from providing the detailed ingredient list in the open response to review. The ingredient list did include other compounds, such as fragrance, that may form SOA upon oxidation. We have modified the manuscript (p. 14, lines 13-16) to mention that these were complex mixtures that were reacted with other potential SOA precursors.*

5. Did the authors also test the composition of the aerosols formed via oxidation of conditioner emissions? I'm curious if the TEM measurements show the same relative elemental composition, or if other peaks (such as carbon) are more abundant. If this spectra is different, then this could point towards the influence of other ingredients on SOA formation.

   *Aerosol collection for microscopy was not performed in the case of the conditioner. Therefore, we cannot comment on the relative contribution from Si-containing and other compounds, and we cannot report the morphology or EDS spectrum. During the Access Review for ACPD, the editor and coauthors agreed that the specific manufacturer and product name would not be released with the article, as the focus on the article was on the general class of personal care products, not specific products. Thus we are refraining from providing the detailed ingredient list in the open response to review. It should be noted that cyclomethicone can refer to other cyclic siloxanes and is not specific to D$_5$. We have modified the manuscript (p. 14, lines 14-15) to state that no microscopy samples were collected for the conditioner source.*

6. Ultimately, I think the authors can do a bit more here to put the personal care product experiments into perspective. As written, a reader could interpret these results to suggest that aerosol formed from personal care products containing D5 is exclusively composed of D5 oxidation products! This is unlikely, of course, and not what the authors are intending to show, so some further discussion should be provided. Furthermore, could the TEM technique be useful in understanding organosilicon aerosol in the atmosphere? Presumably, most particles containing silicon in modes < 100 nm would have been derived from secondary processes. Could these TEM measurements be a useful tool in identifying aerosols resulting from organosilicon oxidation?

   *Thank you for your suggestions. Our intentions were to not say the generated SOA is exclusively the result of D$_5$ but merely to show we do get SOA from oxidizing personal care products, and those products (at least for anti-perspirant) have similar SEM EDS spectra as that with pure D$_5$ SOA. We have revised the manuscript appropriately (p. 14, lines 13-16). We think the SEM-EDS (or TEM-EDS) technique might be a useful method of finding these particles in the environment. But we feel that readers will make that inference on their own, and have not directly stated it.*

7. Page 15, lines 3-11. In this section, the authors discuss the relevance of cVMS SOA formation and potential shortcomings of this work. The authors rightfully demonstrate that cVMS will contribute a small fraction to total SOA; however, I believe the authors are overextending when they try to frame these results into the measurements conducted by Bzdek et al. (2014). Specifically, I take issue with the extrapolation of the

organosilicon SOA content. The nano-particle measurements conducted by Bzdek et al. provide a constraint to nucleation processes close to the source, but downwind of major cities like L.A., chemical and meteorological conditions could change and limit the oxidation of organosilicon precursors. Furthermore, an extrapolation to 1.5 ug/m3 seems quite unrealistic. That amount of SOA is comparable to the background SOA in LA believed to come from regional biogenic sources (~2 ug/m3, Hayes et al. 2015).

I recommend that the authors refrain from extrapolating to suggest that there could be unexplained sources of organosilicon precursors. I think it is sufficient to cite Tang et al. (2015) and McDonald et al. (2018) to show that organosilicon SOA precursors are emerging as an important source of VOCs in urban areas, and that incorporating this work into models will help to constrain the VCP impact on urban SOA formation.

*While we are personally interested to reconcile that large amount of Si found by Bzdek with the low average atmospheric burden that we anticipate, we are happy to remove that from the paper (p. 16, line 33 – p. 17, line 5), as it is probably too speculative. Thanks.*

**Other Comments**

Page 2, Lines 8 - 10. This sentence seems a bit out of place here, and doesn't necessarily requires its own paragraph. I suggest moving it elsewhere in the introduction ( perhaps after the last sentence at line 32?)

*Moved to page 2, lines 25-27.*

Page 3, Lines 1-20. This material would be better presented as a discussion rather than as an introduction. I would recommend moving this to section 3.1

*Moved to Section 3.1, page 10 lines 3-22.*

Page 3, Line23-24. This sentence reads as if concentration, size, and morphology were also measured by EDS. Please rephrase.

*Revised on page 3, lines 24-25.*

Page 4, Line 6. Please add "by photolysis of water" after "in situ"

*Done on page 4, line 10.*

Page 4, Line 11. It would be clearer to say "temperature-controlled"

*Done on page 4, line 17.*

Page 4, Line 14. It would be clearer to the reader if you mention that cyclomethicone and cyclopentasiloxane often refer to the same molecule.

*Done on page 4, lines 21-22.*

Page 5, Line 5. Please note that the "D5 water bath temperature" translates to variations in precursor concentrations.

*Combined first and second sentences to make the sentence clearer on page 5, lines 15-16.*

Page 5, Lines 11 - 12. From my understanding, it's not crucial that the concentrations be the same; rather, it's important that the OH reactivity be low enough that losses due to OH titration can be ignored. What concentration range of D5 and SO2 was being injected into the OFR?

*We included this statement to acknowledge OH suppression caused by external reactants detailed in the PAM chamber chemical kinetics literature (Li et al., 2015; Peng et al., 2016; Peng et al., 2015). In the chamber, OH exposure is dependent on residence time, RH, light intensity, and external OH reactivity (reactant concentration multiplied by the OH rate constant, units of $s^{-1}$). Higher external OH reactivity leads to lower OH exposure in the chamber, therefore characterizing the OH exposure using higher $SO_2$ loadings compared to $D_5$ can result in underestimated $D_5$ OH exposure. We have revised this statement (p. 5, lines 23-25) to make this clearer and have added discussion of the implications to this in the results and discussion (p. 11, lines 21-27).*

*It is important to note that we are below the recommended limit of external OH reactivity (30 $s^{-1}$) for all experiments so that OH concentrations are not suppressed to the extent that other non-OH chemistry is favored.*

*Incoming $SO_2$ concentrations were 450 – 1200 ppb [1.1 x$10^{13}$ – 2.9 x$10^{13}$ molec $cm^{-3}$] (p. 6 line 4)*

*Incoming $D_5$ concentrations were 290 – 740 μg $m^{-3}$ [4.7 x$10^{11}$ – 1.2 x$10^{12}$ molec $cm^{-3}$] (p. 11 line 28)*

*Accounting for OH rate constants, $SO_2$ external OH reactivity loadings were 5.4 – 28 times higher than $D_5$.*
   *$D_5$: 0.7 – 1.9 $s^{-1}$*
   *$SO_2$: 10.1 – 26.3 $s^{-1}$*

Page 5, Line 21. The term "disappearance" suggests that the SO2 was lost by some unknown process. I would recommend replacing with "…measurement of reacted SO2 in the OFR".

*Done on page 6, lines 3-4.*

Page 5, Line 25. Please add "constant" after "rate".

*Done on page 6, line 8.*

Page 6, line 2. How well do the OH exposures calculated from Eqns 1 and 2 agree?

*The OH exposure calculations are compared on page 11, lines 15-20.*

Page 6, Line 26. The semi-colons should be replaced with commas.

*We have gone through the manuscript and replaced semicolons with commas where appropriate.*

Page 7, Line 12. I would refer to "the DMT" or the "DMT Instrument" as a "CCN counter"

*We have gone through the manuscript and replaced "DMT" with "CCN counter" where appropriate.*

Page 7, Line 12-13. Consider removing the phrase "The controlled variable in the DMT instrument" and replace with "The thermal gradient was varied from…" for brevity.

*Done on page 7, line 26.*

Page 7, line 20. By "source", do you mean from the DMA monodisperse outlet?

*Yes, rephrased accordingly on page 8, lines 3.*

Page 7, line 27 - Page 8, Line 10. The details of data QA are not needed here. I suggest removing, or placing in the supplemental information.

*This section has been moved to the supplement under Section S7: Hygroscopicity.*

Section 2.5. I'm somewhat confused by the thermal degradation experiments. The authors don't explain in detail why these experiments were performed, or why these might be relevant in the atmosphere. Are there cases in which D5 might be exposed to high temperature that it would decompose and form lower volatility components? Or, is this a check to evaluate potential biases in the volatility experiments? If it is the latter, I would suggest moving this discussion to Section 2.4 and refer to these experiments as controls.

*Our motivation was to determine if the chemicals could degrade or be oxidized due to high temperature heating leading to low volatility products. This motivation is referenced in Section 3.6 of the Results and Discussion. An additional statement has been added to page 16, lines 3-5 explaining that these tests also serve as a volatility control test.*

Page 10, line 21. What do the authors mean by "reasonable ways?"

*As expected. Sentence has been changed on page 11, lines 30-31.*

Page 10, line 23. It seems to me that the experiment reached steady state long before 20 hrs! No need to change - I'm just impressed by the length of the experiment.

Page 10, line 24 - 25. This sentence is a bit akward. I might consider rephrasing as "This was followed by a sampling period, during which four gas samples were taken to determine D5 SOA yields."

*Rephrased as suggested on page 12, lines 2-3.*

Page 10, line 31. I'm confused why the authors implicate the D5 injection rate as the primary cause for variability. Is it a mixing issue?

*We were not sure what was causing the variability, we had some indications that the water level of the water bath containing the heated leg of Teflon (containing liquid D5) may have been causing variable injection rates. We also suspected that the lights may vary in power in a periodic fashion.*

Page 13, line 19. Please add "that" between "shows" and "no". Also, it is incorrect to write "ka *observed* smaller particles" - please choose a different verb.

*Thanks, changed accordingly on page 15, line 8.*

Page 13, lines 20-22. Can the authors please clarify the last statement about the higher oxidation state of smaller particles? Is that because the larger particles contain more semivolatile components (e.g. dimers, or second-generation monomers?).

*Zhao et al. (2015) and Winkler et al. (2012) report that smaller particles tend to have higher hygroscopicity than larger particles. They suggest smaller particles tend to have more oxidized composition due to lower volatility (Kelvin effect) and also possibly due to higher surface area/volume impacting the incorporation of later generation products or heterogenous oxidation. This section has been revised in the manuscript on page 15, lines 9-13.*

Section 3.5, Figure S13. The 10 nm mode is quite noisy. If it is difficult to quantify, why include this mode in the figure? I would find this figure more useful if I could see the zoomed-in traces for 20-110 nm experiments.

*Revised figures S13 and S14 in the Supplement excluding 10 nm particles.*

Section 3.6. Would thermal degradation be a significant process influencing the lifetime of D5? If so, please provide some justification. If not, I feel like these experiments are better characterized as volatility control experiments - i.e., to ensure that changes in particle concentrations during volatility experiments are not the result of residual gaseous D5 that could degrade and lead to particle formation.

*We viewed this not as an issue of lifetime, but as potential indoor source of inhalable materials. If contact of D5 to hot surfaces (i.e. hairdryer in a bathroom after use of personal care products; cooking elements; heating elements) led to any aerosol formation, then this could be a relevant indoor source. We have changed the section title to Thermal Aerosol Production.*

[revised manuscript text omitted]